# communications

## psychology

# Emotional event perception is related to lexical complexity and emotion knowledge

Zhimeng Li [1✉], Hanxiao Lu [2], Di Liu[3], Alessandra N. C. Yu [4] & Maria Gendron [1✉]

Inferring emotion is a critical skill that supports social functioning. Emotion inferences are typically studied in simplistic paradigms by asking people to categorize isolated and static cues like frowning faces. Yet emotions are complex events that unfold over time. Here, across three samples (Study 1 $N = 222$; Study 2 $N = 261$; Study 3 $N = 101$), we present the Emotion Segmentation Paradigm to examine inferences about complex emotional events by extending cognitive paradigms examining event perception. Participants were asked to indicate when there were changes in the emotions of target individuals within continuous streams of activity in narrative film (Study 1) and documentary clips (Study 2, preregistered, and Study 3 test-retest sample). This Emotion Segmentation Paradigm revealed robust and reliable individual differences across multiple metrics. We also tested the constructionist prediction that emotion labels constrain emotion inference, which is traditionally studied by introducing emotion labels. We demonstrate that individual differences in active emotion vocabulary (i.e., readily accessible emotion words) correlate with emotion segmentation performance.

[1] Department of Psychology, Yale University, New Haven, Connecticut, USA. [2] Department of Psychology, New York University, New York, NY, USA. [3] Department of Psychology, Johns Hopkins University, Baltimore, MD, USA. [4] Nash Family Department of Neuroscience, Icahn School of Medicine at Mount Sinai, New York, NY, USA. ✉email: zhimeng.li@yale.edu; maria.gendron@yale.edu

How do we come to infer what others are feeling? This skill, widely known as emotion perception or emotion recognition, is often studied as if emotions are objects that can be recognized from isolated facial expressions or other non-verbal behaviors[1]. This *object perception framework* not only has intuitive appeal for navigating our social worlds but would provide a powerful basis for applications within fields like healthcare[2], consumer behavior[3], artificial intelligence[4] and law enforcement[5]. Here, responding to accumulating evidence that the object perception framework lacks strong empirical support[6], we pursue an alternative framework for assessing emotion perception: as a form of event perception[7]. This *event perception framework* better captures individual variation and cultural consensus in the emotions that are inferred from naturalistic emotional cues, which are embedded in context. Further, studying emotions as events can reveal how individual differences in language and accessible knowledge about emotions shape our inferences about emotions in others.

Accumulating evidence on the generation and perception of emotional cues is inconsistent with the dominant object perception framework. First, this object perception framework is motivated by the assumption that there is a robust mapping between facial actions and inner emotional experiences. Yet people only rarely generate predicted patterns of facial expressions during instances of emotion[6,8], and there is remarkable diversity in the range of facial cues that are used to convey the same emotion[9]. For example, if there was robust mapping between the internal emotional state of anger and scowling, studying how people perceive isolated scowls would be sufficient to understand how people perceive anger. However, there are many instances when an individual could infer anger that is not captured by a paradigm where a scowling expression is presented to a perceiver. The empirical evidence does not imply that there is no signaling via the face or other channels in emotional episodes but instead implies that the signaling is complex and variable. Second, perceivers appear to rely heavily on contextual cues when inferring emotion, including physical contexts like scenes[10], and what a target is doing with their body[11–13], information about the unfolding situation[9,14,15] and the meanings conveyed by what others say[16]. This implies that the face rarely acts as a simple object of perception in everyday life, in contrast to how psychology experiments are typically conducted. Third, emotion perception is strongly constrained by language. Not only is emotion perception performance sensitive to the language embedded in tasks[17,18], but other context effects are most robust when they invoke language, suggesting that the concepts that words anchor serve to constrain perceptions[19,20]. Together, these findings reveal that the object perception framework insufficiently accounts for the nature of emotion perception.

Previous research examining the contextual nature of emotion inference is still largely characterized by paradigms rooted in the object perception framework, in which researchers rely on trial-based designs with single cues as targets. This critique includes prior research focused on language as a form of top-down constraint[21–24]. Further, provided emotion words in forced-choice designs may serve as guardrails or distractors that result in inflated or deflated performance compared to designs such as free-labeling[25]. Extant paradigms that do measure emotion perception dynamically (and thus not as static objects) have limited contextual cues (e.g., the Empathic Accuracy Paradigm[26,27]) or introduce emotion labels as response choice (e.g., the Inferential Emotion Tracking Paradigm[28]). These task features hence may limit our ability to observe individual variation in active emotion vocabularies (i.e., the emotion words that are generated by the individual). Studying individual differences in active emotion vocabularies may lend insight into how emotion perception proceeds outside of the confines of laboratory settings in more naturalistic and complex contexts.

Studying emotions as events can enhance the ecological validity of existing approaches. Emotions (and our perceptions of them) can be considered events because they unfold over time, are constrained by the context and by preceding emotions[14,29,30], and are causally nested with constituent sensory components[31]. The existing research that considers temporal unfolding of emotions reveals that perceivers are highly sensitive to the dynamics of non-verbal behavior. Dynamic facial movements (compared to static) lead to greater agreement in emotion inferences[32], as well as distinct and more pronounced patterns of neural activation, including in regions involved in face processing[33,34]. People can also reliably rate the dynamics of another person's affective state over time[35,36] and can identify the onset of and transitions between emotions in videos of emotionally expressive targets[37,38]. Further, perceivers hold mental representations of these dynamics. Perceivers within the same cultural group agree on the complex temporal dynamics of how people's faces move when they are feeling different emotions[39] and how one emotion transitions into another[40–42].

Perceivers also understand emotions as causally nested within chains of events. By as early as the second year of life, young children have expectations about which emotional expressions will occur given the circumstances[43]. These expectations can even be imposed onto neutral or completely masked expressions such that preceding emotional context impacts judgments[14,28,44,45], gaze[45] and neural activity[46] in adult perceivers. Together, these findings suggest that perceivers treat emotions like events such that they are sensitive to their temporal dynamics and causal embedding.

Empirical[47] and theoretical[7] developments in the study of event perception can therefore be extended to the study of emotion perception dynamics. Event perception theory[7,48,49] proposes that people mentally represent events unfolding in a spatiotemporal framework as "working event models" such that people use a priori knowledge about the structure and components of events to make predictions. When incoming sensory features misalign with predictions, this is hypothesized to increase prediction errors and result in an update of the working event model. When the event model is updated, an event boundary occurs, which gives rise to the subjective experience of a novel event.

There are strong parallels between this account of event segmentation and emotion perception, where evidence suggests that emotion concepts serve as a source of predictive top-down influence[50,51]. In the case of emotion, increases in prediction error due to a poor match between conceptually driven expectations and sensory input (e.g., facial expressions, body movements, narratives) may produce emotion boundaries, signaling the initiation of a new instance of emotion. Emotion words may serve as a particularly efficient means of bringing these conceptual expectations online[24,52] which in turn shape emotion inferences[53,54]. Indeed, evidence suggests that making emotion concept knowledge more accessible via language priming can lead to earlier visual awareness[55], faster detection of the onset of new expressions[56] and earlier N400 responses following congruent emotion words, which are reflective of facilitated emotion categorical expectancies[57]. Preceding emotion words can even provide visual bias toward emotionally congruent facial expressions followed, leading to enhanced visual awareness and attention to the congruent faces[55].

Here, we pursue the top-down effect of language on emotion perception in more naturalistic dynamics. We developed the Emotion Segmentation Paradigm (Fig. 1) by drawing from several existing tasks. Building on the recently developed Inferential

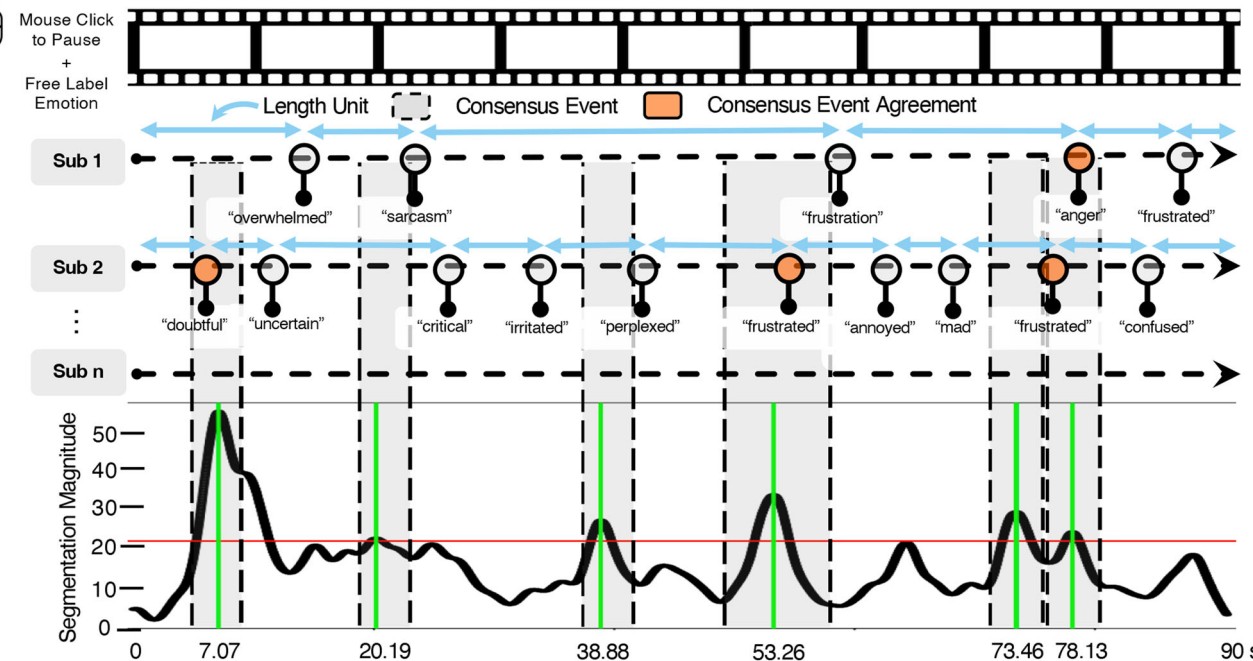

**Fig. 1 Overall framework of the Emotion Segmentation Paradigm.** In the Emotion Segmentation Paradigm, participants pause (unshaded round dots) to freely provide an emotion label or phrase. Length Unit (blue double sided arrows) is the length between two consecutive pauses. Segmentation magnitude reflects how many individuals identified an event at a given point in time and is generated based on group data. Consensus events (gray shaded zones with black dotted line boundaries) are identified as windows around time points where a significant number of people indicated an emotion (solid green lines, which represent local maxima). An individual's data is then compared to these consensus events. Pauses within the consensus event windows are considered as consensus event agreement (orange shaded round dots).

Emotion Tracking (IET) Paradigm[28], in which participants continuously tracked emotions in dynamic stimuli, we ask participants to view emotionally evocative film clips and to indicate when they perceive changes in any of the target character's emotions (what we refer to as an emotion segmentation). A strength of this approach is that it approximates real-life spontaneous emotion perception by utilizing contextually-rich film clips that depict a variety of emotions as stimuli[58]. Not only do we examine consensus in emotion segmentations, similar to the Inferential Emotion Tracking Paradigm[28], but we also innovate by examining individual differences in adherence to this group-level consensus[59,60]. Consensus is a powerful approach because it reflects an individual's fit with their cultural group(s)[61–64] and can be beneficial in domains where there is no clear accuracy criterion[65]. Our focus on group consensus therefore also distinguishes the paradigm from the Empathic Accuracy Paradigm[26,27], which allows participants to indicate observed emotions, but focuses on emotional displays within dyadic or single narrator settings and evaluates performance based on agreement with the target person only. While this is a valuable approach, it is also important to measure whether individuals infer emotion in a culturally typical manner, which may reflect a more generalized skill to infer emotion (i.e., across various relational partners).

We further build on the Inferential Emotion Tracking Paradigm[28] and some versions of the Empathic Accuracy Paradigm[26,27] by asking participants to provide their own labels for the emotions they infer, rather than providing a set of target words for them to choose from. This allows us to examine the active emotion vocabularies of participants. Such a design circumvents artificial accessibility of conceptual knowledge driven by emotion category labels[66].

Lastly, we draw on analytic frameworks from the event segmentation literature[67], including metrics to quantify how finely or coarsely grained an individual's segmentations are[68] and how to identify group consensus on segmentation boundaries[69,70].

Using this paradigm, we report on three studies (one pre-registered) with a total of 584 participants that examine whether there are meaningful individual differences in the ability to segment emotions and whether these individual differences are related to active emotion vocabularies.

## Methods

**Study 1 Participants.** The study was approved by the Yale University Institutional Review Board (IRB #: 2000026863). Participants provided informed consent to participate in the study by reading the consent form and selected the option to proceed with the study in Qualtrics. We recruited 241 participants via Prolific (native English speakers, born and currently living in the United States, none of which has failed both attention checks) and removed 19 low-effort participants who did not successfully segment all videos to ensure unbiased computation of paradigm metrics. This process resulted in a final sample $N = 222$ ($M_{age} = 32.18$, $SD_{age} = 9.82$; 114 men, 98 women, 10 participants who identified as non-binary; 145 identified as White, 22 identified as Black or African American, 19 identified as Asian, 11 identified as Hispanic or Latinx, 3 identified as American Indian or Alaskan Native, 21 identified as mixed race, and 1 identified as 'other'.) The gender and racial identification were provided by participants as responses to demographic questions. Participants were paid the minimum hourly wage of the State of Connecticut ($13/h) at the time of data collection. On the level of subjects, we also used the outliers_mad function (b = 1.4826, threshold = 3) in the Routliers package to detect outliers for each of the five Emotion Segmentation Paradigm metrics, the two lexical complexity metrics, as well as the four questionnaire scores

(as described below). The outliers_mad function uses the Median Absolute Deviation Method to detect outliers based on deviation from the median of the residuals, ensuring robustness of detection for data that is not normally distributed. This resulted in 40 total data points being removed, spanning across 40 participants and 11 metrics.

**Study 1 Stimuli**. To select stimuli, we screened all film clips from Chen and Whitney[14], and selected 6 affective clips from 6 movies: *Me Before You* (75 s), *Before Sunset* (174 s), *Love Rosie* (127 s), *Selena* (179 s), *Spotlight* (69 s), and *Where the Heart Is* (141 s). This subset of clips was selected based on (1) minimal cinematic editing, and (2) frequent emotional changes in the characters. We verified that each clip contained a number of emotional changes in the initial pilot with a larger set of 9 clips and $N = 20$ participants. We also chose 3 non-affective clips as control stimuli from different scenes from some of the same films: *Before Sunset* (115 s), *Selena* (49 s), and *Spotlight* (45 s). Each clip involved people performing daily activities (i.e., going upstairs, opening the door, hanging a coat).

**Study 1 Procedure and measures**. The study was conducted in the context of discovery without preregistration. The study was conducted online via Qualtrics and a custom-built platform across two sessions which were spaced at least 1 week apart. In the first week, participants were randomly assigned to complete either the task session or the questionnaire/demographic session. One week after the completion of the first session, participants were invited back to complete the other session. The interval between the two sessions was designed to minimize any possible spill-over effects between sessions (i.e., questionnaires containing emotional scenarios and language may activate relevant conceptual knowledge that may interfere with the performance of emotion segmentation, or vice versa).

In the task session, participants first completed an audio test to ensure they would have full audio access to the videos and were then directed to the task. The Emotion Segmentation Paradigm contained 2 practice trials and 9 task trials, the latter included 6 affective clips and 3 non-affective clips in separate blocks, counterbalanced across participants. In the affective block, for each trial, the video was played on the left half of the page, and the right half of the page displayed the instruction "Pause the video whenever you think there is a change in EMOTION in any of the characters." Participants clicked on the "Pause" button below the instruction to pause the video, typed words or phrases describing the emotions they have identified in the text box that appeared and clicked on the "Resume" button to resume the video. Participants were encouraged to use a phrase if they did not have access to an emotion label given previous work suggesting that people make meaning of emotional expressions in variable ways, including describing antecedents and action (tendencies)[71]. In the non-affective blocks, the process was the same with the following exceptions: Participants were instructed to "Pause the video whenever you think there is a change in BEHAVIOR/ACTIVITY in any of the characters." Participants were not asked to provide words or phrases to describe behaviors or activities, consistent with the procedure used in the event segmentation literature.

Before starting on each block, participants completed a practice trial and could only proceed with three or more times of segmentation (i.e., pause the video and label emotions), while those segmented less than three times were required to attempt again. Those who failed the practice trial the second time were not allowed to proceed. Participants who passed the practice trial were directed to the formal task trials, which were presented in

the same format as the practice trials. After completion of all 9 task trials, participants were instructed to return to the Qualtrics page for debrief, where they indicated whether they had watched any of the film stimuli and the strategies they have applied during segmentation (for details on debrief see Supplementary Note 2).

In the questionnaire session, participants completed the following four questionnaires in randomized order. (1) The Geneva Emotion Recognition Test–Short Version (GERT-S) is an emotion perception task featuring dynamic videos where actors portray emotions with facial expressions and tonal expressions while speaking nonsensical phrases[72]. Participants were introduced to 14 emotion words and their corresponding definitions and completed two practice trials. For each trial, participants viewed the video and then selected which one of the 14 emotions the actor was demonstrating. Participants completed 42 trials in total. The overall score was computed by taking the raw count of the number of correctly identified trials, which was automatically calculated and aggregated in the Qualtrics task. The measurement demonstrated good reliability from Study 1 ($\omega_T = 0.80$) and Study 2 ($\omega_T = 0.54$). (2) The Situational Test of Emotion Understanding–Brief (STEU-B) is a validated measure of situational knowledge about emotions from the emotional intelligence literature[73]. Participants completed 19 multiple-choice questions, where they read the description of a situation and chose from the 5 given emotions the one that was most likely to result from the given situation. For each question, the selection of the correct emotional response resulted in the score of 1, while the selection of any other options resulted in the score of 0. The overall score was calculated as the sum of all trial scores. The measurement demonstrated adequate reliability for both samples from Study 1 ($\omega_T = 0.65$) and Study 2 ($\omega_T = 0.54$). (3) The Psychological Well-Being Scale (PWB) is a 42-item scale that measures six aspects of wellbeing and happiness (Autonomy, Environmental mastery, Personal growth, Positive relations with others, Purpose in life, and Self-acceptance)[74]. We calculated the scores for each of the six subscales, which we then aggregated into the overall score of PWB (Study 1: $\omega_T = 0.96$; Study 2: $\omega_T = 0.97$). We were specifically interested in the Positive Relations with Others subscale (Study 2: $\omega_T = 0.84$; Study 2: $\omega_T = 0.82$) and used this subscale score for subsequent concurrent validity testing. We expected that dynamic emotion perception should be associated with positive social relationships[75], reflected as a positive correlation between the Emotion Segmentation Paradigm metrics (described below in the "Data Analysis" section) and the Positive Relations with Others subscale scores. Participants indicated level of agreement with 42 statements (e.g., "In general, I feel I am in charge of the situation in which I live.") using a 7-point Likert scale (1 = strongly agree; 7 = strongly disagree; 20 items were reverse coded). (4) The Autism Spectrum Quotient–10 Items (AQ-10) is a scale commonly used as a screening tool for Autism Spectrum[76]. Participants indicated the level of agreement ("strongly agree", "slightly agree", "slightly disagree", "strongly disagree") with 10 statements. For four items, we coded the selection of "Definitely Agree" or "Slightly Agree" as the score of 1 and the other two options as the score of 0. The rest of the items were reverse-coded. The overall score is the sum of all trial scores ($\omega_T = 0.55$). This measure was included to allow for a robustness check, as we expect the metrics to demonstrate high reliability within the subsample who scored less than 6 on this screener tool. More details are further described in Supplementary Note 4.

At the end of the STEU-B questionnaire, an attention check item was inserted which read "Sometimes people do not read the questions carefully. Can you select distress as the option for this question?" A similar attention check item was inserted at the end of the PWB questionnaire which read "Sometimes people do not read the questions carefully. Please select 'strongly agree'."

Participants who failed both attention checks were excluded from the study.

In addition to the two-session main study, we also invited 167 participants back to complete a conceptual similarity task[77] as an additional, exploratory measurement of individual differences in conceptual knowledge of emotion. Participants rated the relatedness of 40 emotional features with respect to 6 distinct emotion categories (anger, disgust, fear, happiness, sadness, and surprise). We then constructed the average Euclidean distance metrics that reflected the conceptual similarity between distinct emotion categories for each individual. These metrics did not demonstrate significant association with the emotion segmentation metrics and were hence not further pursued in Study 2. Detailed description of the task, analysis and results can be found in Supplementary Note 5.

**Study 1 Data Analysis**. Data was collected in three waves spanning from March to August 2021. All data have been formally tested for normality and equal variance. We computed five paradigm metrics in total from the affective trials of the Emotion Segmentation Paradigm. Three metrics were computed to quantify individual differences in dynamic emotion perception, namely the Mean Length of Unit (MLU), Segmentation Agreement (SA), and Consensus Event Agreement (CEA). Each metric was computed for each participant for each affective video stimulus and scores were then averaged across stimuli to result in a single score representing each metric. Two metrics were computed to quantify individual differences in active emotion vocabulary, namely the Number of Unique Labels (NUL) and Semantic Agreement Score (SAS). Each metric is described in detail next.

Mean Length of Unit (MLU) is a measure of how fine-grained versus coarse-grained one's segmentations are and is based on the segmentation grain in the event segmentation literature[78]. Mean Length of Unit was calculated by dividing the length of each stimulus by the number of segmentations (i.e., keypresses pausing the video). A larger Mean Length of Unit score indicates a more coarse-grained segmentation characterized by longer intervals between two segmentations on average[48]. A lower score indicates a more fine-grained segmentation characterized by shorter instances identified between two segmentations.

Segmentation Agreement (SA) is a measure drawn from the segmentation literature[67] that reflects how similar an individual's pattern of segmentation is to the normative pattern of segmentation by the group. Each video stimulus was divided into bins of 1 s, which were then coded as "1" if the participants segmented during that bin and "0" if otherwise. This binary distribution was summed across participants, resulting in a time series of segmentation density on the group level. Segmentation Agreement was calculated by computing the point-biserial correlation between the individual's segmentation time series to that of the normative group time series, scaled based on the number of segmentations the individual. Segmentation Agreement values ranged from 0 to 1, with 0 being the lowest possible level of agreement and 1 being the highest.

Consensus Event Agreement (CEA) is a metric newly developed for the current study, similar to Segmentation Agreement, but is robust to slight deviations in the timing of segmentation keypress relative to the group. In the Segmentation Agreement metric, being early or late to segment an event compared to the group norm can similarly lead to a lower score. Instead, the Consensus Event Agreement metric reflects whether individuals segmented the same instances of emotion as the group. For each video stimulus, we first used the segmag package in R[79] to calculate the segmentation magnitude for each

participant across time by centering a Gaussian distribution ($SD = 0.8$)[70] around each keypress. If multiple distributions overlapped for a participant, the maximum value was used to ensure an equal weight of each participant on the group-level segmentation magnitude. The segmentation magnitudes corresponding to each participant at each time point were then summed up into the overall segmentation magnitude across time, resulting in the group-level segmentation magnitude of each video stimulus. The higher group-level segmentation magnitude at a certain time point indicates the greater number of participants segmented around that time point. We then conducted bootstrapping of the group-level segmentation magnitude distribution with 500 iterations and a critical probability value of 0.95 to define the threshold for statistically significant segmentation magnitude. We then used the function get_eb_times to identify the time points with segmentation magnitude exceeding the threshold. These time points were then considered as timestamps of consensus events of the video. We merged consensus events that were within 800 ms to allow for potential individual differences in reaction time to press pause[70]. Critical events identified within 800 ms may be attributed to individual differences in normative reaction time rather than reflecting actual group consensus on the emotion event. The timestamps of the merged consensus events were the average of the timestamps of the overlapping consensus events.

We then used the changepoint package in R[80,81] to calculate the time window around the timestamp of each consensus event. We use the function cpt.meanvar to detect time points with statistically significant changes in mean and variance centering the group-level segmentation magnitude at the timestamps using the PELT method. For each consensus event, we computed the differences between the consensus event timestamp and these time points and identified the time points with the maximum negative and minimum positive differences relative to the consensus event timestamp. These two timepoints then served as the upper and lower values of the window of said consensus event. For consensus events, we compared the time points of segmentation made by participants with the window of consensus events. If the time point of segmentation made by participants fell within the window of a consensus event, we considered participants as having identified said consensus event. Consensus Event Agreement was calculated as the proportion of consensus events identified by a participant across all 6 affective video stimuli.

Number of Unique Labels (NUL) quantifies unique emotion labels generated by an individual within a given video. It was calculated by taking the raw count of unique labels generated by each participant for each of the 6 affective video stimuli and averaging these values. Using the tm package in R, we removed stopwords (which include the common English stopwords in the tm package but also customized stopwords identified from the raw data), extra blank spaces, numbers and punctuations from the body of labels generated by each participant, and lemmatized the labels so that the variant forms of the same label were counted as one label (e.g., "anger" and "angry" will be considered as one distinct label). We then extracted the number of unique emotion labels as well as their corresponding frequencies from the cleaned body of labels.

We computed several metrics to quantify the semantic meaning of the labels that participants generated. Using the same set of labels used to generate Number of Unique Labels, for each event, we quantified the semantic meaning of the label(s) generated by a given participant using the AffectVec[82] word embedding space, which provides embedding scores of over 70,000 words with about 200 emotion categories. For the first metric, we focused on the within-person differentiation of

emotion concepts across trials based on the loading scores of the labels generated on a set of discrete emotions. We constructed the semantic profile for each emotion and correlated the distinct emotion semantic structures with each other, yielding within-person between-emotion metrics. We refer to these metrics collectively as the Semantic Distinctiveness Score (SDS). See Supplementary Note 3 for a full description of the calculation of the Semantic Distinctiveness Score.

For the second metric, we compared individual semantic loadings to a group-level semantic structure based on the loadings of each label generated for consensus events and averaged across all label loadings for that event producing a group-level semantic profile for each event. We then correlated individual semantic structure for a given event with the group-level semantic structure. We refer to this metric as a Semantic Agreement Score (SAS). See Supplementary Note 3 for a full description of the calculation of Semantic Agreement Score.

In addition to the active emotion vocabulary metrics of the Emotion Segmentation Paradigm, we also analyzed the emotion labels generated by participants using the NLP tool TAALES, which examines a body of words (in this case the unique set of labels generated by each participant) and produces scores corresponding to various lexical features[83]. We selected from TAALES output the parameters reflecting the familiarity and age of acquisition of the unique labels generated. Familiarity scores are based on how familiar the word is to adults. Age of acquisition is the age at which the word is learned.

**Study 2 Participants**. The study was approved by the Yale University Institutional Review Board (IRB #: 2000026863). Participants provided informed consent to participate in the study by reading the consent form and selected the option to proceed with the study in Qualtrics. We aimed to recruit $N = 260$, the sample size at which the effect size of correlations stabilize[84]. A sensitivity analysis showed that we were statistically powered to detect the effect size of 0.214 ($a = 0.008$ for multiple comparison, power = 0.8), which fell below the range of previous effect sizes of Study 1 ($rs = 0.26 - 0.49$), suggesting we were sufficiently powered. We oversampled to account for data loss. We recruited 273 participants (native English speakers, born and currently living in the United States, none of which failed both attention checks) and removed 12 low-effort participants who did not segment all stimuli, resulting in a final sample size of $N = 261$ ($M_{age} = 35.24$, $SD_{age} = 11.64$; 111 men, 140 women, 9 participants who identified as non-binary or others; 194 identified as White, 17 identified as Black or African American, 11 identified as Asian, 12 identified as Hispanic or Latinx, 23 identified as mixed race, and 4 identified as 'other'). The gender and racial information were provided by participants as responses to demographic questions. Participants were paid the minimum hourly wage of the State of Connecticut ($13/h) at the moment of data collection.

**Study 2 Stimuli**. To our knowledge, there are no available repositories of documentary clips that are suitable for studying emotion perception dynamics. We hence constructed a set of stimuli from documentaries (see Supplementary Note 1 for the full process of stimuli construction). To maximize the level of naturalness and spontaneity of the stimuli, we looked for observational documentaries[85] that center on human subjects and attempt to capture everyday life in an unobtrusive manner. Additionally, to control for possible interfering effects of language and culture, we restricted our search to documentaries shot in the U.S. using English as the primary language. We ultimately arrived at a set of 9 clips from 7 documentaries with the average length of each clip at 96.65 s. The clips portray a variety of daily interactions occurring under multiple settings, including dinner gatherings, conversation between doctor and patients, reunion with family members, etc. The age range and ethnic/racial identity of the people featured were also more diverse than those featured in Study 1 stimuli (see Supplemental Note 1 for observed demographic information of the people featured in Study 2 stimuli).

**Study 2 Procedure and measures**. The study was preregistered (preregistration available here: https://aspredicted.org/84G_GQB). Study 2 followed the same general procedure and design as Study 1 with the following modifications. First, as described above, participants segmented documentary clips and there was no non-affective block. For the emotion segmentation instructions, the example word "rude" was replaced with "happy" since rude was not strictly an emotional word but instead described as an affective trait. As in Study 1, participants completed a questionnaire session, which included a streamlined set of measures including the GERT-S ($\omega_T = 0.64$), STEU-B ($\omega_T = 0.54$) and the newly added twenty-item Toronto Alexithymia Scale (TAS-20)[86] ($\omega_T = 0.90$). TAS-20 is a 20-item measurement with three sub-scales (Difficulty Describing Feelings, Difficulty Identifying Feeling, Externally-Oriented Thinking). Participants indicated the level of agreement using a 5-point Likert scale (1 = strongly disagree, 5 = strongly agree; 5 items were reversely coded). The total score is the sum of responses to all 20 items. In alignment with the scaling direction of other questionnaires, the level of agreement was arranged in descending order (i.e., from strongly agree to strongly disagree) in Qualtrics rather than in ascending order, as per original scale. This reordering did not affect the coding of scores.

**Study 2 Data analysis**. Data was collected in four waves spanning from December 2021 to February 2022. All data have been formally tested for normality and equal variance. Study 2 followed the same process of computation and compilation of Emotion Segmentation Paradigm metrics, lexical complexity metrics (using TAALES) as well as questionnaire coding as Study 1. We also conducted the same outlier removal process as Study 1, which resulted in 58 total data points being removed, spanning across 58 participants and 13 metrics. Similar to Study 1, we computed the Omega across trials for Consensus Event Agreement and Number of Unique Labels to examine the reliability of the metrics.

**Study 3 Participants**. The study was approved by the Yale University Institutional Review Board (IRB #: 2000026863). Participants provided informed consent to participate in the study by reading the consent form and selected the option to proceed with the study in Qualtrics. We aimed to recruit $N = 101$, the sample size which, according to the power analysis, is sufficiently powered to detect the estimation of an ICC of 0.7 (Precision = 0.1; $a = 0.05$, power = 0.8). We oversampled $N = 120$ to account for attrition. There were no low-effort participants removed for not segmenting all stimuli nor participants who failed both attention checks. After accounting for attrition, the final sample was $N = 101$ ($M_{age} = 34.94$, $SD_{age} = 11.69$; 53 men, 43 women, 5 participants who identified as non-binary or others; 67 identified as White, 11 identified as Black or African American, 5 identified as Asian, 5 identified as Hispanic or Latinx, 11 identified as mixed race, and 2 identified as 'other.') The gender and racial information were provided by participants as responses to demographic questions. Participants were paid the minimum hourly wage of the State of Connecticut ($13/h) at the moment of data collection.

**Study 3 Stimuli**. Study 3 used the same set of stimuli as Study 2 (i.e., 9 clips selected from documentaries). See Study 2 Method section and Supplemental Note 1 for details on the stimuli and stimulus construction.

**Study 3 Procedure and measures**. In Study 3, participants completed the Emotion Segmentation Paradigm two times with at least 1 week interval in between. Demographic information was collected at the end of the first week. No questionnaire session was included as was in Study 1 and Study 2 since Study 3 concerns mainly with the test-retest reliability of the Emotion Segmentation Paradigm. The study was not preregistered.

**Study 3 Data analysis**. Data was collected in three waves spanning from May to June 2022. All data have been formally tested for normality and equal variance. We computed the paradigm metrics (i.e., Mean Length of Unit, Segmentation Agreement, Consensus Event Agreement, Number of Unique Labels, Semantic Agreement Score) for each subject's performance each week, resulting in two sets of measurements for each subject. We computed the intraclass correlation coefficient[87] following the recommendation by Mcgraw and Wong[88]. Given the temporal order of the two sets of measurements and our interest in the absolute agreement between the two values of each measurement, we selected the type of ICC suitable for two-way models that focus on single measurement. This type corresponds to the ICC type "ICC2" in the ICC function output of the psych R package. In addition, we also computed the Pearson correlation ($a = 0.05$) between two sets of measurements as a supplementary measure for test-retest reliability.

**Reporting summary**. Further information on research design is available in the Nature Portfolio Reporting Summary linked to this article.

## Results

**Experimental data overview**. In Study 1, participants ($N = 222$) completed our Emotion Segmentation Paradigm in which they watched 6 affective film clips and paused whenever they perceived a change in any of the characters' emotional states, even if they did not have a term/label immediately in mind. At these pauses, they were then instructed to label the emotion with a word or phrase. Participants also viewed 3 non-affective (control) film clips and paused/labeled when there was a change in the activity of the target, akin to a classic event segmentation paradigm. We collected the time points of segmenting pauses and corresponding labels using our custom portal (Supplementary Fig. 1), from which we computed metrics reflecting emotion segmentation performance and active emotion vocabulary. In Study 2 ($N = 261$), we replicated Study 1 using naturalistic stimuli extracted from documentaries to examine the generalizability of the paradigm (for details on stimuli construction see Supplementary Note 1). We also focused our analysis on a subset of metrics (Consensus Event Agreement, Number of Unique Labels and Age of Acquisition, as described below and in detail in "Methods") and followed our preregistered plans (preregistration available here: https://aspredicted.org/84G_GQB) with deviations or exploratory analyses noted below. In Study 3 ($N = 101$), we collected a separate sample to examine the test-retest reliability of the paradigm metrics using the Study 2 stimuli (i.e., documentary clips).

**Individual differences are captured by the emotion segmentation paradigm**. To examine individual differences in emotion segmentation, we computed the Mean Length of Unit (MLU),

Segmentation Agreement (SA) and Consensus Event Agreement (CEA) from task data obtained using affective film clips (Fig. 1). Both Mean Length of Unit and Segmentation Agreement were borrowed from the event segmentation literature. Mean Length of Unit indexes the average length of segmented units, which reflects how finely or coarsely grained an individual's segmentations are. Segmentation Agreement captures how an individual's time series of segmentation matches that of the group[67,78]. Consensus Event Agreement was a new metric we developed to avoid the penalty that Segmentation Agreement places on earlier detection of the onset of emotions relative to the group peak. Using the segmag R package[79], we identified for each video a number of group-level consensus events that are time points at which a significant number of people paused to indicate emotion perception and constructed around each consensus event the time windows using changepoint detection[80,81]. Correct identifications of consensus events were based on any segmentation that occurred within the event window. Based on the group-level consensus events, we also computed the Sentiment Agreement Score (SAS) and the Sentiment Distinctiveness Score (SDS) of the labels each individual produced. Semantic Agreement Score captured the (graded) agreement between the sentiment expressed by the participant's label and the average sentiment of the group using the AffectVec word embedding space[82]. Leveraging the same word embeddings, the Semantic Distinctiveness Score captured the within-person differentiation between different emotion concepts. We also computed the Number of Unique Labels (NUL) which also reflected the lexical richness of the generated labels (see "Methods" section for details on metrics development).

Overall, the results suggest that the Emotion Segmentation Paradigm reliably captured individual differences in dynamic emotion perception performance. To test the internal reliability of the paradigm metrics in Study 1, we computed the Omega total scores[89] for the three emotion segmentation metrics (i.e., Mean Length of Unit, Segmentation Agreement, Consensus Event Agreement) of both the affective and the non-affective blocks, and for Number of Unique Labels of the affective group only. Most of the main paradigm metrics, Mean Length of Unit ($\omega_T = 0.9$ for affective stimuli, 0.73 for non-affective stimuli), Consensus Event Agreement ($\omega_T = 0.85$ for affective stimuli, 0.79 for non-affective stimuli) and Number of Unique Labels ($\omega_T = 0.95$ for affective stimuli) demonstrated strong internal reliability. Segmentation Agreement did not yield as strong a reliability ($\omega_T = 0.63$ for affective stimuli, $\omega_T = 0.60$ non-affective stimuli), suggesting it may not be a robust individual difference measure to quantify dynamic emotion perception skill. Semantic Agreement Score similarly demonstrated at best moderate reliability ($\omega_T = 0.59$ for affective stimuli), suggesting its low robustness (for details on exploratory analysis and computation of Semantic Agreement Score, see Supplementary Note 3). The results were replicated in Study 2, with Mean Length of Unit ($\omega_T = 0.89$), Consensus Event Agreement ($\omega_T = 0.8$) and Number of Unique Labels ($\omega_T = 0.94$) demonstrating good internal reliability, and Segmentation Agreement ($\omega_T = 0.61$) and Semantic Agreement Score ($\omega_T = 0.50$) demonstrating weaker reliability.

Results further show that all metrics demonstrate adequate test-retest reliability (Mean Length of Unit: ICC(100) = 0.740; $P < 0.001$; Segmentation Agreement: ICC(100) = 0.552; $P < 0.001$; Consensus Event Agreement: ICC(100) = 0.662; $P < 0.001$; Number of Unique Labels: ICC(100) = 0.702; $P < 0.001$) except for the Semantic Agreement Score (ICC(100) = 0.336; $P < 0.001$). Calculation of the Pearson correlation to further evaluate test-retest reliability yields similar results (Mean Length of Unit: $r(99) = 0.750$; 95% CI, (0.67, 1.00), $P < 0.001$; Segmentation Agreement: $r(99) = 0.560$; 95% CI, (0.44, 1.00), $P < 0.001$; Consensus Event Agreement: $r(99) = 0.663$; 95% CI, (0.56,

1.00), $P < 0.001$; Number of Unique Labels: $r(99) = 0.711$; 95% CI, (0.62, 1.00), $P < 0.001$; Semantic Agreement Score: $r(99) = 0.344$; 95% CI, (0.19, 1.00), $P < 0.001$. Exploratory analysis on the Semantic Agreement Score data was conducted with regard to its low test-retest reliability. Results show that the sentiment agreement of the labels produced by each individual was adequately correlated, suggesting that the low test-retest reliability might be due to multi-level aggregation (for details see Supplementary Note 3).

To examine relationships between all task metrics in Study 1 (see Supplementary Fig. 5 for correlations between all paradigm metrics), Bonferroni corrected pairwise correlations within emotion segmentation metrics (Mean Length of Unit, Segmentation Agreement, Consensus Event Agreement) were conducted. Results show that the emotion segmentation metrics significantly correlated with one another (Fig. 2a, also see Supplementary Fig. 2). The strong positive correlation between Consensus Event Agreement and Segmentation Agreement (Spearman's

$\rho(219) = 0.730$; 95% CI, (0.66, 0.80), $P < 0.001$) suggests that both metrics capture reliable variance in emotion segmentation performance, with both metrics similarly reflecting how the segmentation pattern of the individual agrees with that of the group. The significant negative correlation between Mean Length of Unit and the other two metrics (Consensus Event Agreement: $\rho(215) = -0.836$; 95% CI, (−0.88, −0.80), $P < 0.001$; Segmentation Agreement: $\rho(216) = -0.409$; 95% CI, (−0.52, −0.29), $P < 0.001$) indicates that as individuals make finer grain segmentations (reflected in smaller Mean Length of Unit), their emotion segmentation performance is also better. We also conducted exploratory analyses to examine the relationship between our emotion segmentation metrics and signal detection parameters of criterion and sensitivity. We found that Mean Length of Unit can be considered a proxy for criterion of segmentation (i.e., the willingness of participants to indicate that an emotional change occurred when it is ambiguous), whereas the other metrics (Consensus Event Agreement, Segmentation

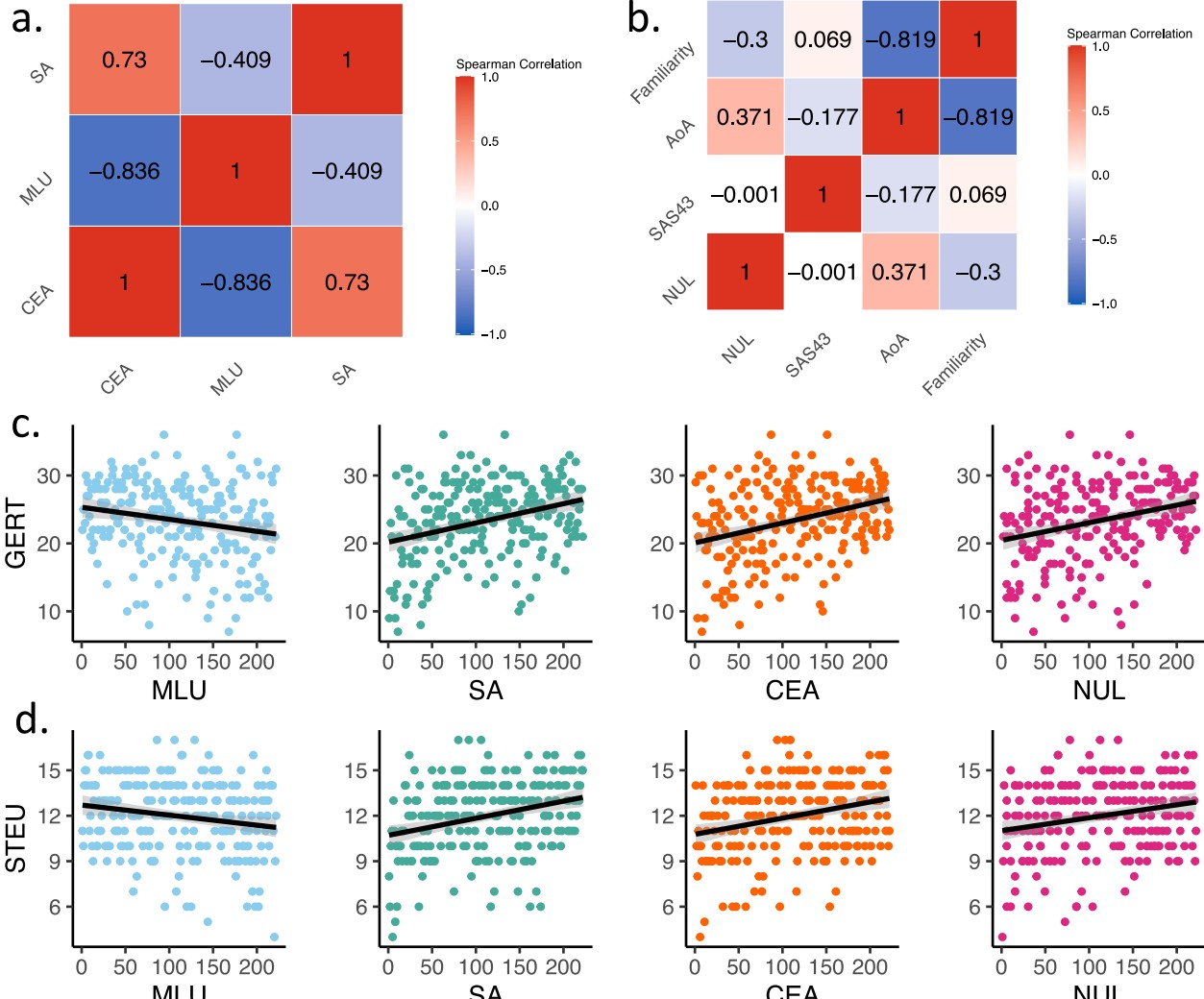

**Fig. 2 Correlation within metrics and between metrics and established measures in Study 1 ($N = 222$). a** The emotion segmentation metrics, Mean Length of Unit (MLU), Consensus Event Agreement (CEA) and Segmentation Agreement (SA), all significantly correlated with one another. **b** The active emotion label metrics, Number of Unique Labels (NUL), Semantic Agreement Score (SAS), Age of Acquisition (AoA) and Familiarity correlated with each other to varying degrees. Semantic Agreement Score did not correlate with any other metrics, while Age of Acquisition and Familiarity correlated with each other, and Age of Acquisition correlated more strongly with Number of Unique Labels. **c, d** While all four paradigm metrics (Mean Length of Unit, Segmentation Agreement, Consensus Event Agreement, Number of Unique Labels) correlated with GERT-S scores in the expected direction, only Segmentation Agreement, Consensus Event Agreement, and Number of Unique Labels were correlated with STEU-B scores. The graphs contain scatterplots of ranked data. For an equivalent figure for Study 2, see Supplementary Fig. 5.

Agreement) relate to both signal detection parameters. For additional details and discussion of these exploratory analyses, see Supplementary Note 6.

We separately examined the relationships between active emotion vocabulary metrics including those previously described (Number of Unique Labels, Semantic Agreement Score) and derived measures of the lexical properties of generated labels, including the average familiarity and average Age of Acquisition (AoA) using the TAALES[83]. We did not include Semantic Distinctiveness Score in the analysis given its exploratory nature (see Supplementary Note 3 for calculation and exploratory analyses involving Semantic Distinctiveness Score). The active emotion label metrics correlated with each other to varying degrees (Fig. 2b, also see Supplementary Fig. 3). First, we found no credible evidence that Semantic Agreement Score correlated with Number of Unique Labels ($\rho(206) = -0.001$; 95% CI, $(-0.13, 0.14)$, $P > 0.999$), Familiarity ($\rho(211) = 0.069$; 95% CI, $(-0.06, 0.27)$, $P > 0.999$) or Age of Acquisition ($\rho(213) = -0.177$; 95% CI, $(-0.30, -0.05)$, $P > 0.999$), calling into question the meaningfulness of this measure. As a result, Semantic Agreement Score was not deemed robust enough to be considered for our focal analysis of the association between active emotion vocabulary and emotion segmentation and is not further reported in the main text (for exploratory analysis of Study 1 data involving Semantic Agreement Score see Supplementary Note 3). In contrast, the lexical complexity variables Age of Acquisition and Familiarity were highly correlated with each other ($\rho(216) = 0.819$; 95% CI, $(-0.87, -0.77)$, $P < 0.001$), in line with the previous literature. Both metrics also correlated in the expected directions with Number of Unique Labels, with Age of Acquisition having a stronger correlation (Age of Acquisition: $\rho(211) = 0.371$; 95% CI, $(0.25, 0.49)$, $P < 0.001$; Familiarity: $\rho(209) = -0.300$; 95% CI, $(-0.42, -0.17)$, $P < 0.001$). Since Age of Acquisition is highly correlated with Familiarity and shared a stronger association with Number of Unique Labels, we only retained Number of Unique Labels and Age of Acquisition as the active emotion vocabulary metrics for our focal analyses in Study 2 testing the relationship between active emotion vocabulary and emotion segmentation performance. Similar to the emotion segmentation metrics, we conducted exploratory analyses to examine the relationship between the active emotion vocabulary metrics and signal detection parameters of criterion and sensitivity. We observed that both Number of Unique Labels and Age of Acquisition correlated only with criterion, but not with sensitivity. For additional details on these analyses, see Supplementary Note 6.

**Convergent validity of emotion segmentation performance**. We examined a number of correlates of task performance (all assessed in a separate session) in the context of discovery in Study 1. The Geneva Emotion Recognition Test (GERT-S)[72] and Situational Test of Emotion Understanding-Brief (STEU-B)[73] were included to test the convergent validity of our emotion segmentation metrics. Positive correlations between the paradigm's emotion segmentation metrics and the scores of GERT-S and STEU-B would indicate good convergent validity of the paradigm, as these measures assess emotion perception performance and situated emotion knowledge, respectively. We computed pairwise Spearman correlations (Bonferroni corrected for multiple comparisons) between each of the paradigm metrics (i.e., Mean Length of Unit, Segmentation Agreement, Consensus Event Agreement, Number of Unique Labels) and the GERT-S (Fig. 2c) and STEU-B scores (Fig. 2d). Results reveal that although all metrics are related to the classic emotion perception measure (i.e., GERT), only Segmentation Agreement, Consensus Event Agreement and

Number of Unique Labels are also related to situated conceptual knowledge of emotion. Segmentation Agreement, Consensus Event Agreement and Number of Unique Labels all positively correlated with the GERT-S score (Segmentation Agreement: $\rho(219) = 0.290$; 95% CI, $(0.16, 0.41)$; $P < 0.001$; Consensus Event Agreement: $\rho(218) = 0.312$; 95% CI, $(0.19, 0.43)$; $P < 0.001$; Number of Unique Labels: $\rho(211) = 0.271$; 95% CI, $(0.14, 0.40)$; $P < 0.001$) and the STEU-B score (Segmentation Agreement: $\rho(220) = 0.291$; 95% CI, $(0.17, 0.42)$; $P < 0.001$; Consensus Event Agreement: $\rho(219) = 0.270$; 95% CI, $(0.15, 0.39)$; $P < 0.001$; Number of Unique Labels: $\rho(212) = 0.199$; 95% CI, $(0.07, 0.33)$; $P = 0.008$). Mean Length of Unit correlated with the GERT-S score ($\rho(216) = -0.205$; 95% CI, $(-0.33, -0.07)$; $P = 0.005$) but we found no credible evidence that it correlated with the STEU-B score ($\rho(216) = -0.122$; 95% CI, $(-0.26, 0.01)$; $P = 0.142$).

To further distinguish between Consensus Event Agreement and Segmentation Agreement, the two metrics that measure segmentation performance, we also conducted an exploratory multiple regression analysis and set up two separate models with GERT-S (Supplementary Table 1) and STEU-B (Supplementary Table 2) scores as outcome variables. Controlling for other metrics, Segmentation Agreement seemed to be the strongest predictor of GERT-S scores ($\beta(205) = 0.27$; 95% CI, $(0.09, 0.45)$; $P = 0.002$). Only Segmentation Agreement significantly predicted STEU-B scores when controlling for other task metrics ($\beta(205) = 0.29$; 95% CI, $(0.11, 0.48)$; $P = 0.002$).

Additional measures included in the context of discovery in Study 1 were (1) the Psychological Well-Being Scale (PWB)[74], (2) a modified version of the conceptual similarity task from Brooks and Freeman[77]. These measures did not yield significant relationships with task performance. Finally, the Autism Spectrum Quotient–10 items (AQ-10)[76] was included as a robustness check on Study 1 results where individuals who may meet criteria for ASD are excluded from analyses (see Supplementary Note 4 for details of analyses using PWB and AQ-10 scores and "Methods" section and Supplementary Note 5 for discussion of the conceptual similarity task).

In the context of confirmation, in Study 2, we found that our primary emotion segmentation metric Consensus Event Agreement was positively correlated with both the GERT-S score ($\rho(255) = 0.292$; 95% CI, $(0.18, 0.40)$; $P < 0.001$) and the STEU-B score ($\rho(244) = 0.257$; 95% CI, $(0.13, 0.37)$; $P < 0.001$). In the context of discovery, we also included in Study 2 the twenty-item Toronto Alexithymia Scale-I (TAS-20)[86], which is a relevant measure for examining the association between-emotion vocabulary and emotion perception[90]. However, we found no credible evidence that Consensus Event Agreement correlated with TAS-20 ($\rho(259) = -0.01$; 95% CI, $(-0.13, 0.11)$; $P = 1.000$) (Supplementary Fig. 4). Note that these Study 2 correlations with Consensus Event Agreement were Bonferroni corrected ($\alpha = 0.0167$) to account for multiple testing.

**The link between active emotion vocabulary and emotion segmentation**. To test our hypothesis that the active emotion vocabulary will be related to segmentation performance, in Study 1, we examined the relationship between the multiple segmentation task metrics (Mean Length of Unit, Segmentation Agreement, Consensus Event Agreement) and each of the two active emotion vocabulary metrics (Number of Unique Labels, Age of Acquisition). Supporting our predictions, results show that all three emotion segmentation metrics were significantly correlated with both active emotion vocabulary metrics (correlations were Bonferroni corrected for multiple testing) (Table 1). There was also some variation in the strength of the association, however. First, Consensus Event Agreement was more strongly correlated

**Table 1 Spearman correlations between emotion segmentation variables (Mean Length of Unit, Segmentation Agreement, Consensus Event Agreement) and active emotion vocabulary variables (Number of Unique Labels, Age of Acquisition) with confidence intervals ($N = 222$).**

| Variable | NUL | AoA |
|---|---|---|
| MLU | −0.86** [−0.91, −0.79] | −0.23** [−0.35, −0.10] |
| SA | 0.41** [0.29, 0.53] | 0.26** [0.13, 0.39] |
| CEA | 0.75** [0.67, 0.82] | 0.27** [0.15, 0.39] |

** indicates $p < 0.01$.

with both active emotion vocabulary metrics (Number of Unique Labels: $\rho(212) = 0.748$; 95% CI, (0.67, 0.82); $P < 0.001$ Age of Acquisition: $\rho(218) = 0.268$; 95% CI, (0.14, 0.39); P < 0.001) compared to Segmentation Agreement (Number of Unique Labels: $\rho(212) = 0.414$; 95% CI, (0.29, 0.53); $P < 0.001$; Age of Acquisition: $\rho(219) = 0.260$; 95% CI, (0.13, 0.39); $P < 0.001$). Furthermore, although Mean Length of Unit correlated more strongly with the active emotion vocabulary metric Number of Unique Labels ($\rho(208) = -0.859$; 95% CI, (−0.91, −0.79); $P < 0.001$), its correlation with Age of Acquisition is the weakest amongst the three segmentation metrics ($\rho(215) = 0.226$; 95% CI, (−0.35, −0.10); $P = 0.002$). Mean Length of Unit only reflects segmentation style (i.e., fine- or coarse-grained) rather than consensus-based emotion segmentation performance, which is more of interest. Among the metrics that do assess consensus-based emotion segmentation performance, Segmentation Agreement predicts both GERT-S and STEU-B scores, suggesting its high convergent validity, but it does not bear the strongest correlation with the active emotion vocabulary metrics. Given our primary hypothesis regarding the relationship between emotion perception and active emotion vocabulary, we highlight Consensus Event Agreement as the primary metric capturing emotion segmentation performance in Study 1 results and focus specifically on this metric in Study 2 to confirm the relationship with the two active emotion vocabulary metrics.

To examine the robustness of the relationship between active emotion vocabulary and emotion segmentation, we conducted a preregistered replication. Study 2 differed from Study 1 in a critical respect: instead of asking participants to segment clips from films, which portray a relatively limited emotional range (based on the free labels provided, see Fig. 3a), we asked participants to segment clips from documentaries, which depicted a greater variety of emotions (Fig. 3b). Segmentation data on this more naturalistic set of stimuli replicated the main finding of Study 1. Consensus Event Agreement positively correlated both Number of Unique Labels ($\rho(251) = 0.666$; 95% CI, (0.59, 0.73); $P < 0.001$) and Age of Acquisition ($\rho(258) = 0.143$; 95% CI, (0.02, 0.26); $P = 0.021$), based on Bonferroni corrected Spearman correlations (Fig. 4). People who have larger and more sophisticated emotion vocabularies are better able to segment the emotions of others, in this case from spontaneous and naturalistic expressions of emotion. We should note, however, that power analysis ($a = 0.025$ for multiple comparison, power = 0.8) suggests we are only powered to detect an effect size as small as 0.189, such that the small effect size between Consensus Event Agreement and Age of Acquisition should be interpreted with caution. (Note that the preregistered analysis plan of our a priori test regarding the relationship between active emotion vocabulary and emotion segmentation contained a typo of $a = 0.25$ instead of $a = 0.025$.)

In the exploratory analyses examining the relationship between our metrics and signal detection metrics, we observed that the

primary emotion segmentation metric, Consensus Event Agreement, and both active emotion vocabulary metrics (i.e., Number of Unique Labels and Age of Acquisition) correlated with segmentation criterion. To examine whether criterion plays a role in the link between Consensus Event Agreement and active emotion vocabulary metrics, we also conducted multivariate regression analysis. Results revealed that neither active emotion vocabulary metrics predicted Consensus Event Agreement when controlling for criterion. For additional details on these analyses, see Supplementary Note 6.

**Generality of segmentation performance.** In order to investigate whether segmentation performance reflects domain-general ability subsuming both event segmentation and emotion segmentation, we further conducted analysis on the three event segmentation metrics (Mean Length of Unit, Segmentation Agreement, Consensus Event Agreement) based on data from non-affective (control) stimuli in Study 1. We first computed the metric-to-metric correlation between the emotion segmentation metrics and event segmentation metrics. Results show that all three event segmentation metrics Mean Length of Unit ($\rho(209) = 0.519$; 95% CI, (0.41, 0.62); $P < 0.001$), Segmentation Agreement ($\rho(220) = 0.223$; 95% CI, (0.10, 0.35); $P < 0.001$) and Consensus Event Agreement ($\rho(219) = 0.425$; 95% CI, (0.31, 0.53); $P < 0.001$) for control stimuli were correlated with their counterparts in emotion segmentation, suggesting that segmentation ability might be domain-general. These domain-general links might also account for the significant correlations (Bonferroni corrected for multiple comparisons) between two of the three event segmentation metrics and GERTS-S (Mean Length of Unit: $\rho(211) = -0.067$; 95% CI, (−0.20, 0.07); $P = 0.491$; Segmentation Agreement: $\rho(219) = 0.228$; 95% CI, (0.10, 0.36); $P < 0.001$; Consensus Event Agreement: $\rho(219) = 0.189$; 95% CI, (0.06, 0.32); $P = 0.007$) as well as between all three metrics and STEU-B scores (Mean Length of Unit: $\rho(212) = -0.152$; 95% CI, (−0.29, −0.02); $P = 0.040$; Segmentation Agreement: $\rho(220) = 0.275$; 95% CI, (0.15, 0.40); $P < 0.001$; Consensus Event Agreement: $\rho(220) = 0.240$; 95% CI, (0.12, 0.36); $P < 0.001$). To further examine if the association between emotion segmentation metrics and the convergent validity measurements is present only due to the shared correlation with event segmentation metrics, we computed partial correlations controlling for the event segmentation metrics. Results showed that, when controlling for the event segmentation counterpart, most emotion segmentation metrics still significantly correlate with GERT-S scores (Segmentation Agreement: $\rho(218) = 0.254$; 95%CI, (0.25, 0.37); $P < 0.001$; Consensus Event Agreement: $\rho(217) = 0.261$; 95%CI, (0.13, 0.38); $P < 0.001$) and STEU-B scores (Segmentation Agreement: $\rho(219) = 0.245$; 95% CI, (0.11, 0.36); $P < 0.001$; Consensus Event Agreement: $\rho(218) = 0.191$; 95%CI, (0.05, 0.31); $P = 0.005$). However, Mean Length of Unit correlates with only GERT-S scores ($\rho(208) = -0.175$; 95%CI, (−0.17, −0.04); $P = 0.02$) but not STEU-B scores ($\rho(208) = -0.059$; 95%CI, (−0.06, 0.08); $P = 0.39$) in the expected direction. Additional correlation analysis (corrected for multiple comparisons) focusing on the primary event segmentation metric Consensus Event Agreement further revealed that domain-general Consensus Event Agreement is correlated with Number of Unique Labels ($\rho(212) = 0.408$; 95% CI, (0.29, 0.52); $P < 0.001$) but found no statistically significant evidence for a correlation with Age of Acquisition ($\rho(219) = 0.118$; 95% CI, (−0.01, 0.24); $P = 0.08$). However, even controlling for the domain-general Consensus Event Agreement, Consensus Event Agreement for emotion segmentation still significantly correlated with Number of Unique Labels ($\rho(211) = 0.697$; 95%CI, (0.62, 0.76); $P < 0.001$), suggesting that emotional segmentation is still uniquely related to active emotion vocabulary. Collectively, these results suggest that while emotion

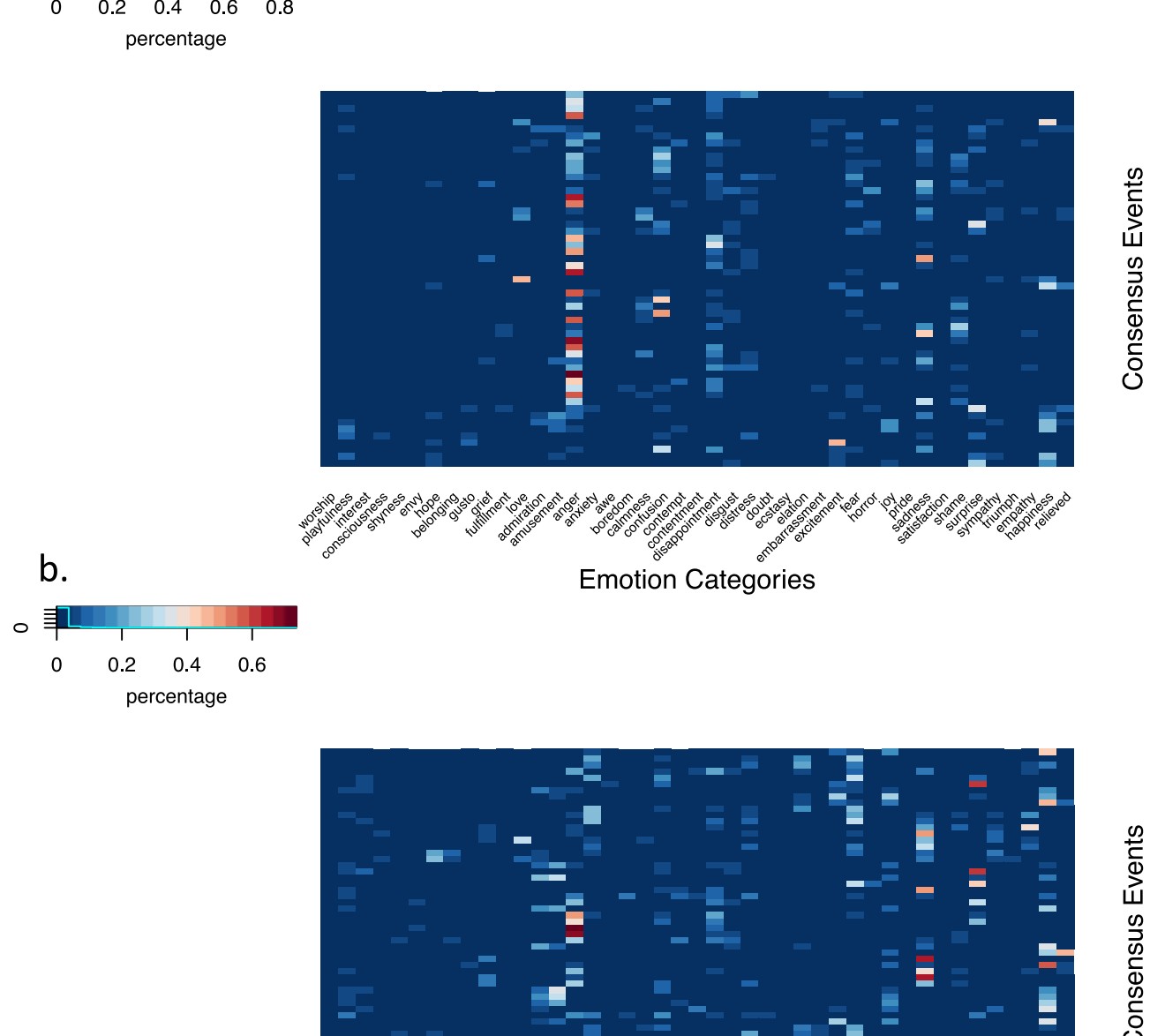

**Fig. 3 Semantic complexity of Study 1 and Study 2 stimuli. a** The film clips from Study 1 disproportionally featured the emotion of anger, as indicated by the high percentage of people identifying anger across consensus events. **b** Documentary clips from Study 2 captured a wider range of emotions, with high percentage of people identifying other emotions such as sadness, surprise and happiness.

segmentation does appear to relate to (and perhaps rest on) the domain-general skill to segment events, the ability to segment emotional events is still uniquely related to validity measures, including STEU-B and GERT-S, as well as active emotion vocabulary. It is hence productive to focus on emotion segmentation ability specifically in replication and extension work.

## Discussion

Across three studies, we demonstrated that the Emotion Segmentation Paradigm can reliably and validly capture individual differences in dynamic emotion perception performance. Our study complements and builds on prior empirical efforts[14,28] by examining individual differences in dynamic emotion perception

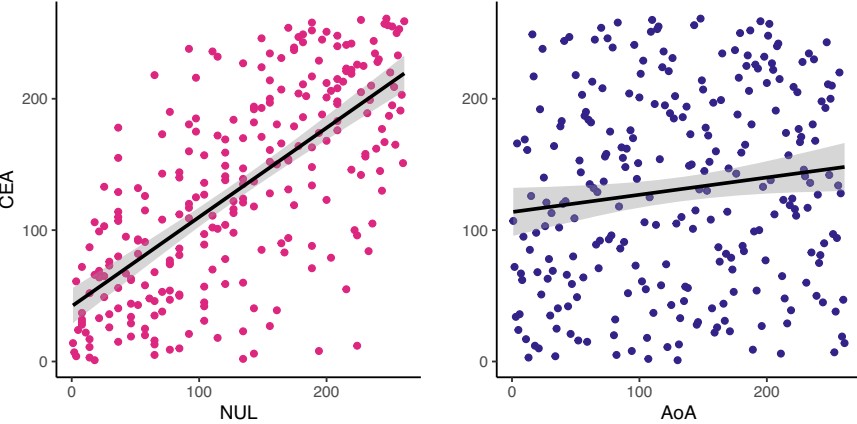

**Fig. 4 Correlation between emotion segmentation metric and active emotion vocabulary metric in Study 2 ($N = 261$).** The primary emotion segmentation metric Consensus Event Agreement (CEA) is positively correlated with both active emotion vocabulary metrics, Number of Unique Labels (NUL) and Age of Acquisition (AoA), suggesting that people with better segmentation performance also possess larger and more advanced emotion vocabularies. See Supplementary Fig. 6 for an equivalent figure in Study 1.

without introducing predetermined emotion category labels. This Emotion Segmentation Paradigm enriches the repertoire of methodological tools available to study emotion perception and individual differences therein. Our approach borrows a number of elements from previous work (i.e., the Inferential Affective Tracking[14] and Inferential Emotion Tracking Paradigms[28]), which assessed intersubject agreement in affect ratings and discrete emotion judgments from film and documentary clips, we also examine consensus in the identification of emotional changes from similar (and some overlapping) stimuli. Chen and Whitney[14,28] found that perceivers rely largely on situational information as cues to make emotion inferences, further stressing the importance of context in emotion perception.

Our work is distinct, however, in adapting analytic methods and theory from the event segmentation literature to investigate individual differences in emotion perception. Specifically, we examined how finely grained segmentations were for individuals and quantified agreement in segmentation timing with others. We found that individuals vary in how finely grained they segmented emotions and in the extent to which their segmentations were temporally synchronous with others.

Importantly, we also examined the role of emotion language in emotion segmentation by collecting freely generated emotion labels from the participants and leveraging tools from natural language processing to quantify the complexity of each participant's active emotion vocabulary. Many standardized emotion stimuli (e.g., NimStim[91]; CEED[92]) are validated through forced-choice tasks, where a separate sample of participants rate the matching between the stimuli (usually facial expressions) and predetermined emotion labels. In contrast, our paradigm constructs group-level semantic consensus from labels generated by participants and evaluates semantic agreement with the group consensus.

We observed and internally replicated the predicted association between active emotion vocabulary and emotion segmentation performance. People who perceive emotion events in higher agreement with the group (manifested in the paradigm as reaching greater emotion segmentation consensus with the group) also generated more unique emotion labels that were obtained at later ages and hence can be interpreted as more semantically complex and sophisticated. These findings complement previous findings from the literature demonstrating a link between emotion labels and emotion perception[17–20]. Critically, the present findings suggest that the impact of emotion language

on emotion perception was not an artifact of previous experimental designs, where researchers introduced labels into the task.

Exploratory analysis using signal detection metrics revealed that both the primary emotion segmentation metric (Consensus Event Agreement) and active emotion vocabulary metrics are correlated with a more liberal criterion to segment. People with a broader and more sophisticated active emotion vocabulary may achieve higher agreement with others by being liberal about what counts as a change in emotion. It is possible that the Emotion Segmentation Paradigm has encouraged a more liberal strategy, given how participants were instructed (i.e., to segment even when they do not immediately know what the emotional change is). Participants may be motivated by the task instruction to "identify" emotional changes regardless of the actual presence of a specific emotional change. Prior research suggests that individuals may adjust their criterion to be more liberal if the structure of the environment makes it more costly to miss a target[93]. While there was technically no cost to missing a target in our paradigm, participants may have interpreted the instructions in such a manner.

The present findings bear the potential to inform educational and clinical practice focusing on improving individuals' emotion perception skill and social functioning. Our findings converge with previous work demonstrating that the acquisition of emotion language facilitates learning of emotional concepts and subsequent emotion perception[94–96]. Admittedly, there are label-independent mechanisms through which emotion concepts may be acquired, such as learning based on a caregiver's statistically regular demonstration of a limited number of emotions[97]. However, training involving emotion language via emotion communication[98] or reading stories with diverse emotion lexicons[99] is still effective in the learning of emotion knowledge and holds promise for improving emotion perception performance[100]. From a basic science standpoint, examining emotion segmentation in developmental samples may supply further evidence for the role of one's active emotional vocabulary in the development of emotion perception ability. The Emotion Segmentation Paradigm can also provide insight into the research centering clinical questions, such as how emotion inferences may vary in people with alexithymia. Alexithymia is characterized by the difficulty in identifying and describing one's own emotion[101] and is associated with language impairment[102]. Correlational and causal neurological evidence also suggests that the brain regions involved in semantic processing are associated with mental

representation of discrete emotions[103]. Hence, our paradigm with its dual focus on emotion language and emotion perception ability may be especially well suited for studying populations with heightened alexithymia.

**Limitations**. The current study has several limitations. For one, the stimuli in Study 1 disproportionally featured anger (Fig. 4). It is possible that the number of anger segmentations observed are not entirely stimulus-driven but based on the accessibility of the anger category for participants. We do observe that in videos where targets are described as "angry", there tend to be more segmentations, on average (see Supplementary Note 7). Additionally, the documentary clips in Study 2 were selected by our research team, rather than randomly sampled from a pre-defined population of clips that were compiled in a perceiver-independent manner. Our standard of selecting videos based on dynamics of emotions featured may have inadvertently resulted in a disproportionate inclusion of videos featuring negative emotions such as anger. Still, it is possible that segmentation for negative emotions is easier, hence skewing the performance of segmentation. This limitation can be addressed with replications and more data-driven and/or theoretically motivated clip selection.

Furthermore, we did not quantify the possible impact of cinematography and editing on people's segmentation of clips. In event segmentation, situational features in narrative films such as the discontinuity in time, space, or spatial movements and actions can predict the identification of event boundaries[104,105]. Although our clips are shorter and often focus on a single spatiotemporal setting, there are cinematography details such as zooming in and switching between focus on faces and the broader situational scene, which may serve as cues for participants to identify emotional instances.

Across studies, we explicitly instructed participants to engage in emotion segmentation. Task-motivated segmentation may vary from spontaneous emotion perception in everyday life. For instance, participants may direct more attention toward the targets expressing emotion than they would normally do in real-life situations. Furthermore, our paradigm diverges from typical event segmentation tasks by asking participants to label at each segment. This introduces discontinuity into the paradigm and may disrupt some aspects of the temporal unfolding of emotional instances. This limitation can be addressed by examining the impact of pausing on the number and timing of emotion segmentations.

Despite the adequate within-subject reliability of the Emotion Segmentation Paradigm metrics, we found Segmentation Agreement to have not as strong a reliability score as the other metrics. Segmentation Agreement was adopted from the prior literature on event segmentation (i.e., domain-general, behavioral events)[7]. Past literature indicates that the reliability we observe is within the published ranges. For example, Kurby and Zacks[67], from which we adopted the calculation of Segmentation Agreement, observed reliability (measured using Cronbach's alpha) ranging from 0.50–0.90. The reliability we observed (Study 1: $\omega_T = 0.63$ for emotion stimuli, $\omega_T = 0.60$ for non-emotion stimuli; Study 2: $\omega_T = 0.63$) fell within this range. However, the within-subject stability of Segmentation Agreement can be sensitive to the specific stimuli being segmented[69]. Furthermore, we did not specify that participants should use either a fine (i.e., segment the video into the smallest units that are meaningful to them) or course-grain (i.e., segment the video into the largest units) of segmentation[106,107]. Our instructions thus may have led to variation, even within individual, in how they approached the task (fine-grained versus coarse-grained) and this may have impacted the reliability of Segmentation Agreement.

Finally, a consensus-based approach used here is limited in that it does not capture other forms of "accuracy" that have been examined in the prior literature studying emotion "recognition".

Whereas some recognition studies use normative ratings of stimuli as ground truth and are thus ultimately relying on consensus, other approaches may use other ground truth criteria. Ground truth may be based on researcher-stipulated stimuli such as facial portrayals that contain specific muscle group movements[108,109], carrying assumptions about clear mappings between these cues and emotions. Ground truth may also be based on the situation in which the cues were elicited[9,110], carrying assumptions about clear mappings between situations and emotions. Finally, ground truth may be based on the self-reported emotion of the target individual[35,111–113], carrying assumptions about individuals' ability and motivation to access and label their emotional states. Each of these approaches has different strengths and weaknesses that should be weighed carefully, including consideration of the evidence supporting their various assumptions.

**Future directions**. Future research using an effectively balanced stimulus set would allow for additional insights into how segmentation behaviors vary by valence (or other affective features). For example, it may be that individuals engage in more fine-grained segmentation for negative compared to positive emotional stimuli. This would fit with prior literature suggesting a negativity bias[114,115] and that negative emotion such as anger sharpens attention[116,117]. This would also allow for the investigation of clinical samples with documented differences in the perception of positive versus negative emotions[118,119].

Future research may also benefit from coding cinematographic manipulations such as cuts and examining if the cinematographic pattern is correlated with the emotion segmentation. In addition, building on the findings of prior work demonstrating context as a strong driver of emotion inferences[28], future research could also further examine whether particular cues such as facial movements or aspects of the context drive segmentations.

Furthermore, for consistency purposes, we have constrained both our sample and the stimuli within the North American cultural and English linguistic context. Future studies may also explore the reliability of the paradigm and generalizability of the current findings with culturally variant samples and stimuli. Moreover, in social settings, people may strategically present emotions different from their actual feelings in the achievement of certain social goals[120,121]. It is hence also of interest to adapt the Emotion Segmentation Paradigm in examining the nuance between perceiving emotion that others overtly expressed and inferring emotion that others experienced. It may be that perceivers agree on expressed emotions, while showing more variability in consensus around inferred felt emotions, for example.

Our exploratory analysis using signal detection metrics demonstrated a positive link between active emotion vocabulary and a liberal criterion for pausing videos. To probe the link between emotion vocabulary and sensitivity to emotion events, future research could use a more classic detection task where the events (and foils) are paused for the participant.

Our paradigm can be further extended to examine emotion segmentation in clinical populations. For example, past research suggests that individuals with depression have a bias toward content of negative valence, which may lead to different perceptual attention[122] and interpretation[123,124] of emotional stimuli. Future research with depressed samples could distinguish between normatively positive, negative, and ambiguous emotional events, which would allow for more precision in measuring what is driving departures from consensus. Furthermore, the current study has focused on the differentiation of others' emotion, rather than one's own emotional experience, which may account for the non-significant correlation between the TAS-20 score and

emotion segmentation performance. Future research may apply the paradigm with instructed focus on one's own emotional experience instead of that of the others.

The Emotion Segmentation Paradigm can also be expanded in future work to more explicitly examine a broader range of labeling behaviors beyond the use of emotion labels. A long-standing observation is that individuals label emotional targets with a range of terms including antecedent conditions and action tendencies, which can be considered constitutive aspects of emotional events[125]. As a result, our focus on emotion labels may underestimate consensus on emotional meanings. Further, cultural groups vary in the tendency to describe emotions using action verbs, bodily state descriptions and mental states descriptions[126,127]. The Emotion Segmentation Paradigm would be a beneficial framework for examining whether these different ways of making meaning (e.g., emotion labels versus action tendencies) impact the grain and relative timing of emotion segmentations.

Future research can also pursue the shared theoretical bases of event and emotion segmentation ability, which may provide insights into the underlying mechanism of emotion labels' involvement in dynamic emotion perception. As outlined earlier, Zacks and colleagues[7,49] propose that segmentation is based on the updating of event models which are deployed in a predictive manner. During emotion perception (conceptualized in the current study as emotion segmentation), active emotion vocabulary may guide efficient access to mental models of emotions that drive multi-modal predictions. Future research could focus on testing this predictive account more directly by manipulating the predictability of incoming sensory inputs in the form of facial expressions, vocal tone, and other verbal or non-verbal cues and examining the corresponding impact on emotion segmentation[51]. Further, it would be interesting to similarly constrain performance by introducing labels, thereby activating a similar emotion vocabulary across individuals, and examine the impact on performance relative to a control condition. Finally, examining the physiological correlates of behavioral segmentation, which may capture spontaneous emotion segmentation during passive viewing, may lend more support for the idea that perceivers are spontaneously perceiving emotion boundaries.

## Conclusion
The conceptualization of emotion perception as dynamic event perception holds the promise of lending new perspectives to both basic science and multiple applied domains in emotion research. Yet relevant research has been constrained by the limited ecological validity of current emotion perception measurements. Borrowing from work in both social and cognitive psychology, we developed and validated the Emotion Segmentation Paradigm, which we also used to replicate the link between emotion perception and emotion label accessibility. We anticipate that the paradigm will enrich the toolkit of emotion perception researchers and enable deeper exploration of the nature of emotion inferences.

## Data availability
The quantitative and qualitative data (deidentified) collected and/or analyzed during Studies 1–3 are available in the OSF repository at https://osf.io/zsu7t/?view_only=90f62ea1056c4b4e995514d99fe59e3a.

## Code availability
The analysis code for Studies 1–3 can be accessed from https://osf.io/zsu7t/?view_only=90f62ea1056c4b4e995514d99fe59e3a.

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

## Acknowledgements

The authors received no specific funding for this work. We thank Da-Jin Chu for developing the custom platform of Emotion Segmentation Paradigm and Calvin Solomon for helping with initial stimulus development.

## Author contributions

M.G. developed the study concept. H.L., A.Y. and Z.L. developed the stimuli. H.L., A.Y. and M.G. designed the initial experiment and planned the analyses. Z.L. and M.G. designed subsequent experiments and planned the analyses. H.L. and Z.L. collected the experimental data. H.L., D.L. and Z.L. analyzed the experimental data. Z.L. and M.G. wrote the manuscript, with contributions from D.L. and edits from H.L. and A.Y.

## Competing interests

The authors declare no competing interests.
