## [Peer Review File · Communications Psychology]

15th Mar 23

Dear Dr Gendron,

Thank you for your patience during the peer-review process. Your manuscript titled "Emotional Event Perception is Related to Lexical Complexity and Emotion Knowledge" has now been seen by 3 reviewers, whose comments are appended below. You will see that they find your work of some potential interest. However, they have raised quite substantial concerns that must be addressed. In light of these comments, we cannot accept the manuscript for publication, but would be interested in considering a revised version that fully addresses these serious concerns.

We hope you will find the Reviewers' comments useful as you decide how to proceed. Should additional work allow you to address these criticisms, we would be happy to look at a substantially revised manuscript. If you choose to take up this option, please highlight all changes in the manuscript text file, and provide a detailed point-by-point reply to the reviewers.

Editorially, we consider it especially important to address the reviewers' methodological concerns, the concerns regarding what conceptual contributions the findings make, and concerns regarding the characterization of the advance the work presents over existing approaches.

We ask you to address the methodological concerns and requests for additional analyses made by Reviewers 1 and 3 (especially those addressing the association between language ability and ability to perform the event segmentation task).

Please elaborate precisely how your study is similar and different to previous research and paradigms, avoiding any novelty claims (describing the work as "novel", "first", "unprecedented", etc.), or mischaracterization of previous work.

Any remaining issues and concerns that limit the scope of your findings should be discussed transparently in a "Limitations" section in the "Discussion".

Please also make sure to include information on IRB approval and informed consent and a working preregistration link as requested by Reviewer 3 that can be accessed anonymously.

Also, please correct the potential typo on page 12, line 363. A correlation is reported as significant but with a high p-value.

If the revision process takes significantly longer than five months, we will be happy to reconsider your paper at a later date, provided it still presents a significant contribution to the literature at that stage.

We understand that due to the current global situation, the time required for revision may be longer than usual. We would appreciate it if you could keep us informed about an estimated timescale for resubmission, to facilitate our planning. Of course, if you are unable to estimate, we are happy to accommodate necessary extensions nevertheless.

Please use the following link to submit your revised manuscript, point-by-point response to the Reviewers' comments with a list of your changes to the manuscript text (which should be in a separate document to any cover letter) and any completed checklist:

[link redacted]

Please do not hesitate to contact me if you have any questions or would like to discuss the required revisions further. Thank you for the opportunity to review your work.

Best regards,

Jennifer Bellingtier

Jennifer Bellingtier, PhD
Senior Editor
Communications Psychology

EDITORIAL POLICIES AND FORMATTING

Editorial Policy: [Policy requirements](https://www.nature.com/documents/nr-editorial-policy-checklist.pdf) (Download the link to your computer as a PDF.)

Furthermore, please align your manuscript with our format requirements, which are summarized on the following checklist:

[Communications Psychology formatting checklist](https://www.nature.com/documents/commsj-psychol-style-formatting-checklist-article.pdf)

and also in our style and formatting guide [Communications Psychology formatting guide](https://www.nature.com/documents/commsj-psychol-style-formatting-guide-accept.pdf) .

* **TRANSPARENT PEER REVIEW:** Communications Psychology uses a transparent peer review system. This means that we publish the editorial decision letters including Reviewers' comments to the authors and the author rebuttal letters online as a supplementary peer review file. However, on author request, confidential information and data can be removed from the published reviewer

reports and rebuttal letters prior to publication. If your manuscript has been previously reviewed at another journal, those Reviewers' comments would not form part of the published peer review file.

* **CODE AVAILABILITY:** All Communications Psychology manuscripts must include a section titled "Code Availability" at the end of the methods section. In the event of publication, we require that the custom analysis code supporting your conclusions is made available in a publicly accessible repository; please choose a repository that provides a DOI for the code; the link to the repository and the DOI must be included in the Code Availability statement. Publication as Supplementary Information will not suffice. We ask you to prepare and upload code at this stage, to avoid delays later on in the process.

* **DATA AVAILABILITY:**

All Communications Psychology research manuscripts must include a section titled "Data Availability" at the end of the Methods section or main text (if no Methods). More information on this policy, is available at <http://www.nature.com/authors/policies/data/data-availability-statements-data-citations.pdf>.

At a minimum the Data availability statement must explain how the data can be obtained and whether there are any restrictions on data sharing. Communications Psychology strongly endorses open sharing of data. If you do make your data openly available, please include in the statement:

We recommend submitting the data to discipline-specific, community-recognized repositories, where possible and a list of recommended repositories is provided at <http://www.nature.com/sdata/policies/repositories>.

If a community resource is unavailable, data can be submitted to generalist repositories such as [figshare](https://figshare.com/) or [Dryad Digital Repository](http://datadryad.org/). Please provide a unique identifier for the data (for example a DOI or a permanent URL) in the data availability statement, if possible. If the repository does not provide identifiers, we encourage authors to supply the search terms that will return the data. For data that have been obtained from publicly available sources, please provide a URL and the specific data product name in the data availability statement. Data with a DOI should be further cited in the methods reference section.

<http://www.nature.com/authors/policies/availability.html>.

REVIEWER EXPERTISE:

Ref #1 social/affective psychology; individual differences

Ref #2 social/affective psychology; emotion expression

Ref #3 social/affective psychology; event segmentation

Reviewer #1 (Remarks to the Author):

This is a very well written manuscript that presents a novel segmentation paradigm. The technique involves playing movies and asking participants to pause the video and freely label the event with a word. The results show high agreement, reliability, individual differences, and associations with language/vocab metrics. The experiments are interesting, the results novel and provocative, and the writing is clear. I have some minor comments and questions, but this is an excellent manuscript that will be of broad interest to the Nature Comms Psych community.

1. Interesting and novel paradigm. Clearly related to the inferential emotion tracking paradigm of Chen et al (PNAS, Cognition, Emotion papers in 2019, 2020, 2021), and the authors here used a small subset of Chen's stimuli, as well. It would be very useful to add a discussion of the similarities and differences between the novel emotion segmentation technique here, and the inferential emotion tracking technique of Chen et al. This should be addressed in the introduction and the discussion section.

2. There are several places in the manuscript with quotes like, "where dynamics of emotion perception, if studied at all, are typically addressed experimentally in a piecemeal fashion." Statements like these aren't quite fair since there are published techniques out there on dynamic emotion perception, including the inferential emotion tracking paradigm, which is continuous and does reveal the dynamics. In fact, that paradigm was used to show that the context is available with a latency that is nearly as short as that for the face itself (Chen, Cognition, 2021, <https://doi.org/10.1016/j.cognition.2020.104549>). Only measuring dynamics allows that kind of temporal resolution. Moreover, the segmentation paradigm proposed here isn't as continuous as implied because it actually introduces discontinuities in the flow of information (pausing the video).

3. The segmentation task involves pausing the videos. Doesn't this in and of itself introduce a kind of discontinuity? Like a second-order discontinuity, for example. Is it possible to control that somehow?

4. What if different observers have different thresholds for what justifies a segmentation? Are there individual differences in that criterion or is that just captured by or assumed to be captured by MLU? The possible issue here is that threshold and criterion are sort of conflated in the MLU metric (MLU could vary because an observer isn't sensitive to the segmentation-relevant information and could also vary because they've set a high criterion for what justifies pausing the video). Maybe there's a way to measure this in signal detection terms (ie, break apart MLU into sensitivity and criterion proxies).

5. Line 185. Is it slightly concerning that SA did not have as strong a reliability score? This seems like a core metric and yet the lower score is odd and would benefit from some interpretation.

6. LL 427-439. As the authors noted, some of the videos have cinematography, they may be staged, and they are often edited (even documentaries can have staging and editing). This may introduce intentional and artificial segmentations. One way to address this in future work would be to measure

context-based segmentation. Some of the videos used here were borrowed from the inferential emotion tracking paradigm, which had separate context-only, actor-only, and context+actor conditions created from the same videos. The cinematography is controlled in this way (e.g., comparing context versus actor) so any difference in segmentation would not be caused by staging but by differences in the availability of segmentation-relevant information. It is not necessary to conduct additional experiments for this manuscript. However, adding this to the future directions portion of this paragraph would be worthwhile because it helps address a potential concern readers will have.

7. Fig. 4 The event segmentation suggests the videos in study 1 disproportionately featured anger, but how do we know this is not caused by the segmentation task itself? Perhaps segmentation is easier for anger? Study 2 videos partially address this, but anger still features quite prominently (the peak cell in Fig 4b is anger, and anger has one of the highest if not the highest average percent). The videos in Chen displayed a broader and more uniform range of emotions (both affect and categorical emotion) as measured by continuous tracking, but only a small subset of those videos was used here, so perhaps the video selection introduced the bias. Is there a way to measure whether anger disproportionately features in virtue of the segmentation paradigm itself?

8. Related to the above. Beyond just “anger” there seems to be a disproportionate prevalence of negative valence in Study 1 videos. Across many negative categories there is higher percent. A split half on positive vs negative affect would probably really highlight this. The source may be the higher salience of negative versus positive events. Additionally, there could also be a normalization problem: what if a video has an overall very positive tone, but is punctuated by changes in emotion. If the mean emotion/affect is, on average, already quite positive/high, then the only salient emotion change there can be is in a negative direction. That will look like many negative segmentations, and imply that the videos contain a lot of anger. That would be incorrect, though: the video is on average very positive but the salient changes can only occur in a negative direction. Study 2 videos seem more balanced on this, but the interpretation of the segmentations remains a bit unclear.

9. Minor: The “Emotion Segmentation Paradigm” is capitalized. The Empathic Accuracy paradigm” is partially capitalized. The “inferential emotion tracking task (paradigm)” is not capitalized. Either capitalize them all, or don’t, but be consistent. Also, be consistent about the use of the word “paradigm” or “task” since those words are interchangeable in all three of these cases.

10. Fig 3. The CEA-NUL correlation is very strong and interesting. What about the possibility that this is due to the threshold idea in point 4 above? That is, one may recognize the need to segment without having the words for it. In that case, a participant might set a different criterion for pausing, but have the exact same sensitivity (MLU might change but that’s an ambiguous metric). This isn’t so much caused by an association between vocabulary and sensitivity-to-segmentation but between language and criterion-for-pausing-videos. A couple of ideas to address this are to play back videos segmented by high vocabulary participants (a kind of ground truth) to those with low language scores. Can those low-vocab individuals nevertheless recognize the correct placement of segmentations (versus, say, lure segmentations in the wrong moments)? This kind of sensitivity could be measured in many different ways, like recollection and familiarity, for example.

Reviewer #2 (Remarks to the Author):

The authors propose a new framework for studying emotion expressions as events rather than objects. Studying emotion perception within a dynamic context and doing so using free labeling is an interesting approach. However, it should be noted that this approach is not precisely new (as the authors note, it has been used in studies on empathic accuracy) nor does it in fact address pertinent problems in current emotion perception research (it may address problems in emotion perception research as it was conducted in the second half of the last century), see also below. The metrics used are also strongly biased towards departures from consensus (both in terms of emotion discrimination as in terms of word use) as the sole individual difference of interest. This limits its applicability for clinical and other contexts where it does not suffice to know that someone does not share the consensus view (which is often a given). Rather than knowing that individuals identified as X do not share consensus on anger expressions it would be more important to know how they perceive these expressions instead.

That said, studying emotions as events is certainly of interest. It is just that it does not provide a better account of emotion expression but rather a complementary account that may be useful for specific research questions for which emotion discrimination ability and consensus perception are of special relevance.

The main rationale for switching the framework for studying emotion expressions is predicated by a – not uncommon – confusion between emotion production and emotion perception. These are not the same process and as such issues that are relevant to emotion production do not have to be relevant to emotion perception. Thus, while it is the case that humans tend to not produce highly stereotypical facial expressions in everyday circumstances, this does not mean that humans do not perceive emotion expressions as reactions to singular events. The use of emotion expressions in the films and literature very nicely demonstrates that most humans perceive emotion expressions as signals of internal states, i.e., emotions, even if it is not clear that they were produced that way.

The authors also grossly overstate the impact of context and prior labeling on emotion perceptions. In the cited study [10] only anger expressions in a disgust context were labelled as disgust rather than anger. The same expressions in sad or fear contexts were still labelled as anger. Given the well-known strong perceptual overlap between anger and disgust this study really only shows a modest contribution of context. The authors also seem to underestimate the proliferation of the use of dynamic and contextualized stimuli in more recent emotion research. In this sense they rightly criticize a research approach that hardly anyone still does anymore.

That available words constrain what can be said is a truism. However, this also applies to the proposed new framework. Hence it is not clear why the new framework presents an advantage on this count.

Given that “standardized emotion stimuli” are usually validated by means of assessing consensus ratings, the use of consensus to assess accuracy is also anything but new. The only difference seems to be that the authors assess consensus on the fly (within their experiment) whereas more typically the consensus was established first, and accuracy then represents – as it does here – the agreement or disagreement with this consensus.

It should also be noted that free labeling is not really a solution to the points with regard to the influence of language. First, people can only use the words they know and if they do not know the

“correct” word, they will surely not be able to use it. Giving labels does not necessarily distort the labeling process as such labels are usually commonly known words, the meaning of which most participants are likely to remember even if they would not produce the word spontaneously.

More importantly, as already shown very nicely by Frijda (1953) when free-labelling people do not necessarily produce emotion words but very often focus on either the antecedents or the action tendencies associated with the expression. In a context, where emotions are viewed as valuable social signals knowing what produced the expression or what behaviors to expect is arguably more relevant than simply affixing a label. This, however, makes it more difficult to assess consensus as some people may use a label (he is angry), others an antecedent (he looks like someone insulted him), and yet others an action tendency (he looks as if he wants to hit someone), all of which may or may not relate to the same emotion concept. The analyses procedure used by the authors is mainly focused on quantifying the use of similar words rather than on actual context and meaning. It is hence unclear to what degree the approach can actually assess emotion decoding – the purported goal.

The metrics used by the authors focus on temporal agreement. That is, to what degree do people agree on a change in observed behavior. This however refers to the ability to discriminate emotions, which is absolutely not the same as recognizing emotions. Only NUL even refers to the content of the labels and there also in a very reduced manner focusing on consensus only. Yet a focus on consensus is actually contrary to the stated rationale which aims to assess constructivist predictions as well. A constructivist prediction would accept that individuals construct meaning based on expression and context – and because this construction is done by individuals’ departures from consensus are as meaningful as is adherence to consensus.

Reviewer #3 (Remarks to the Author):

This manuscript proposes a novel emotion labeling during video viewing. The reliability of the paradigm was tested by calculating the level of consensus across participants in how they labeled different moments. They found a substantial level of agreement in participants' segmentation pattern. In addition to the inter-participant reliability, in a study (study 3), test-retest reliability was tested. The validity of the paradigm was tested by looking at the relationship between well known, established emotion sensitivity and recognition measures and new emotion labeling measures obtained using the current paradigm. Using these emotion labeling measures, the research reports new findings too; they tested for an effect consistent with the constructionist theory of emotion. For example, individuals that emotionally labeled events more similar with all other individuals (those high on consensus agreement) had richer emotion vocabulary.

There has been an increase in attention on people's continuous processing of social and affective information. This new approach proposed here is a good addition, especially for those looking at the role of language and conceptual knowledge in perception. The current research has other admirable features: (1) preregistration (but it cannot be accessed using the link in the manuscript-this should be fixed in the revision if there is an opportunity for revision...), (2) analyzing data additionally obtained using a less emotional, 'control' video, and (3) replications across multiple studies using different stimuli. Overall, the manuscript has potential to make a contribution to the field of affective science and social psychology, but revisions are needed to fully realize the potential. I

follow with some mostly straightforward, non-conceptual notes.

1. "We also focused our analysis on a subset of metrics ... followed our pre-registered plans (available here: https://aspredicted.org/see_one.php)" (line 152):

I cannot access the preregistration. To look up a preregistration on AsPredicted, I need the AsPredicted number and the last name of the/an author. I do not see the number.

2. "The age range and ethnic/racial identity of the people featured were also more diverse than those featured in Study 1 stimuli" (line 691):

It would be useful to know how specifically diverse the people were (e.g., apparent races, actors' races/ages).

3. " On the level 486 of subjects, we also used the outliers_mad functon($b = 1.4826$, threshold = 3) in the Routliers package to detect outliers for each of the five Emotion Segmentation Paradigm metrics, ..." (line 485)

It would be useful if you could at least conceptually explain how the outliers are determined using outliers_mad function? Since a lot of key points in the manuscript are about the consensus in emotion judgments and you do not want to appear as if you were removing people with idiosyncratic (yet valid) emotion perception pattern from further analysis.

4. We merged consensus events that were within 800ms to account for the delay in reaction time to press pause70. The timestamps of the merged consensus events were the average of the timestamps of the overlapping consensus events. (line 620)

Why 800ms? A rationale would be useful.

5. There are many acronyms. Some of them are introduced in the text without the full spelling. Even after a couple of reads, I find myself confused. This issue stood out to me in particular in figures; sometimes it has a label that I cannot fully understand even when I refer to the main manuscript (e.g., "SAS43" in Figure 2b). It would be useful to have acronyms written at the end of every figure legend for those that are not familiar with the paradigm.

6. It would be useful to have an equivalent figure to Figure 2 for study 2 as well. That will help the audience gauge the rigor of the main effects found here.

7. Likewise, it would be useful to see Figure 3 equivalent for study 1. (2.4 The Link Between Active Emotion Vocabulary and Emotion Segmentation) I believe the effect was repeated for CEA.

8. The way why Positive Well Being Scale and Autism Spectrum Quotient were included in the questionnaire series was too ambiguous in my opinion ("We were specifically interested in the Positive Relations with Others subscale ... and used this subscale score for subsequent concurrent validity testing."). It might be useful to briefly describe the respective prediction (what positive or negative relationship you predicted, between each of these measures and the measure from your paradigm)—just one short sentence each perhaps.

9. "These RSA metrics ..." (line 574) this may be confusing to the readers unfamiliar with RSA (also, the acronym should be spelled out previously) because what RSA is and how it is applied in this

context is not clear. It would be useful to either (1) briefly explain how it was done, or (2) completely lose the acronym ("RSA") so the authors as well as readers do not have to worry about it. Either case, you could simply refer to Suppl Note 5.

Note: New text in provided quotes is in blue font and old text is in black font.

Comments from Reviewer #1

1. Interesting and novel paradigm. Clearly related to the inferential emotion tracking paradigm of Chen et al (PNAS, Cognition, Emotion papers in 2019, 2020, 2021), and the authors here used a small subset of Chen’s stimuli, as well. It would be very useful to add a discussion of the similarities and differences between the novel emotion segmentation technique here, and the inferential emotion tracking technique of Chen et al. This should be addressed in the introduction and the discussion section.

We thank the reviewer for their comment. They are correct at identifying the relationship between our paradigm and the Inferential Emotion Tracking Paradigm (IET for below). We now briefly described the IET paradigm and its main strength. The IET paradigm (Chen & Whitney, 2020) and the Emotion Segmentation (ES for below) Paradigm are similar in that both make use of dynamic stimuli (i.e., movie clips) and examine consensus in the identification of emotions. Differing from the IET paradigm, we collect spontaneously generated emotion labels rather than discrete emotion judgments based on a predetermined set of categorical labels. Further, our analytic approach draws more heavily on the event segmentation literature, including the characterization of individual differences in segmentation (and the correlates of such individual differences). These differences are illustrated in the discussion section on page 14, lines 428-434 as below:

“Building on previous work (i.e., the Inferential Affective Tracking¹ and Inferential Emotion Tracking Paradigms²), which assessed intersubject agreement in affect ratings and discrete emotion judgments from film and documentary clips, we also examine consensus in the identification of emotional changes from similar (and some overlapping) stimuli. Chen and Whitney^{1,2} found that perceivers rely largely on situational information as cues to make emotion inferences, further stressing the importance of context in emotion perception.

Our work is distinct, however, in adapting analytic methods and theory from event segmentation literature to investigate individual differences in emotion perception. Specifically, we examined how finely grained segmentations were for individuals and quantified agreement in segmentation timing with others. We found that individuals vary in how finely grained they segmented emotions and in how well their segmentations were temporally synchronous with others.

Importantly, we also examined the role of emotion language by collecting freely generated emotion labels from the participants, and leveraging tools from natural

language processing to quantify the complexity of each participant's active emotion vocabulary. ”

2. There are several places in the manuscript with quotes like, “where dynamics of emotion perception, if studied at all, are typically addressed experimentally in a piecemeal fashion.” Statements like these aren't quite fair since there are published techniques out there on dynamic emotion perception, including the inferential emotion tracking paradigm, which is continuous and does reveal the dynamics. In fact, that paradigm was used to show that the context is available with a latency that is nearly as short as that for the face itself (Chen, Cognition, 2021, <https://doi.org/10.1016/j.cognition.2020.104549>). Only measuring dynamics allows that kind of temporal resolution.

The reviewer is correct in pointing out that paradigms such as IET have significantly contributed to empirical investigation into dynamic emotion perception, and are especially useful at examining key aspects of emotion perception such as use of context. Please see our response to the previous comment (Reviewer 1, comment 1) to see how we have better characterized the similarities and differences between the IET paradigm and our own. We have removed the quote referred to in the comment and tempered language surrounding the lack of prior studies examining emotion inference in a dynamic fashion (for example, on page 13, lines 423-426, page 2, lines 58-66, page 3, line 71-72, page 14, line 449-454 as shown below).

“Across three studies, we demonstrated that the Emotion Segmentation Paradigm can reliably and validly capture individual differences in dynamic emotion perception performance. Our study complements prior empirical efforts^{1,2} by examining individual differences in dynamic emotion perception without introducing predetermined emotion categorical labels. ”

“Previous research examining the contextual nature of emotion inference is still largely characterized by paradigms rooted in the object perception framework, in which researchers rely on trial-based designs with single cues as targets...Extant paradigms that measure emotion perception dynamically (and thus not as static objects) have limited contextual cues (e.g., the Empathic Accuracy Paradigm^{3,4}) or introduce emotion labels as response choice (e.g., the Inferential Emotion Tracking Paradigm²). ”

“Studying emotions as events will help to contribute to enhancing the ecological validity of existing experimental approaches.”

“We also observed and replicated the predicted association between active emotion vocabulary and emotion segmentation performance. People who perceive emotion events

in higher agreement with the group (manifested in the paradigm as reaching greater emotion segmentation consensus with the group) also generated more unique emotion labels that were obtained at later ages, and hence can be interpreted as more semantically complex and sophisticated.”

3. Moreover, the segmentation paradigm proposed here isn't as continuous as implied because it actually introduces discontinuities in the flow of information (pausing the video). The segmentation task involves pausing the videos. Doesn't this in and of itself introduce a kind of discontinuity? Like a second-order discontinuity, for example. Is it possible to control that somehow?

This is an important consideration that the reviewer points out. We agree that our paradigm does introduce discontinuities by asking participants to label at the segmentation points. We have added the following language to the Limitations section of Discussion (page 15, lines 512-516):

“Furthermore, our paradigm diverges from typical event segmentation tasks by asking participants to label at each segment. This introduces discontinuity into the paradigm and may disrupt some aspects of the temporal unfolding of emotional instances. This limitation can be addressed by examining the impact of pausing on the number and timing of emotion segmentations.”

4. What if different observers have different thresholds for what justifies a segmentation? Are there individual differences in that criterion or is that just captured by or assumed to be captured by MLU? The possible issue here is that threshold and criterion are sort of conflated in the MLU metric (MLU could vary because an observer isn't sensitive to the segmentation-relevant information and could also vary because they've set a high criterion for what justifies pausing the video). Maybe there's a way to measure this in signal detection terms (ie, break apart MLU into sensitivity and criterion proxies).

This comment is particularly thought-provoking and prompted us to conduct some additional analyses to examine how underlying signal detection parameters relate to our metrics based on the event segmentation literature.

To explore, we computed signal detection theory metrics (i.e., sensitivity and criterion). We did so by considering consensus event windows as “signal trial” (hit) and the time intervals between consensus event windows as “noise trial”. Thus, a segmentation within a signal trial window would be considered a *hit*; no segmentation within a signal trial would be considered a *miss*; a segmentation within a noise trial would be considered a *false alarm*; no segmentation within a

noise trial would be considered a *correct rejection*. We computed SDT parameters based on the formulas published in Macmillan & Creelman⁵. We adopted loglinear adjustment⁶ to account for extreme cases when the hit count was zero (i.e., participants did not identify any consensus event).

In both Study 1 and Study 2, results demonstrate that MLU is not related to sensitivity. Overall, we also found that perceivers have relatively low sensitivity (i.e., discrimination between consensus events and non-events). On the other hand, a more liberal criterion is strongly positively associated with lower MLU (Spearman's ρ ranged between .92 and .96). These results suggest that MLU can be considered a proxy measure for criterion, but not sensitivity.

These results are reported in Supplementary Note 6 and references briefly in the Results section on page 7, lines 228-234, as follows:

“We also conducted exploratory analyses to examine the relationship between our emotion segmentation metrics and signal detection parameters of criterion and sensitivity. We found that MLU can be considered a proxy for criterion of segmentation (i.e., the willingness of participant's to indicate that an emotional change occurred when it is ambiguous), whereas the other metrics (CEA, SA) relate to both signal detection parameters. For additional details and discussion of these exploratory analyses, see Supplementary Note 6.”

5. Line 185. Is it slightly concerning that SA did not have as strong a reliability score? This seems like a core metric and yet the lower score is odd and would benefit from some interpretation.

We agree with the reviewer that the low reliability of Segmentation Agreement (SA) warrants further interpretation and discussion. To address this comment, we have edited the manuscript to include the following discussions on pages 15-16, lines 517-530:

“Despite the adequate within-subject reliability of the Emotion Segmentation Paradigm metrics, we found SA to have not as strong a reliability score as the other metrics. SA was adopted from the prior literature on event segmentation⁷ (i.e., domain-general, behavioral events) and past literature on event segmentation suggests that the reliability we observe is within the published ranges. For example, Kurby and Zacks⁸, from which we borrowed the calculation of SA, observed reliability (measured using Cronbach's alpha) ranging from 0.50-0.90. The reliability we observed (Study 1: $\omega_T = 0.63$ for emotion stimuli, $\omega_T = 0.60$ for non-emotion stimuli; Study 2: $\omega_T = 0.63$) fell within this range. However, the within-subject stability of SA can be sensitive to the specific stimuli

being segmented⁹. Furthermore, we did not specify that participants should use either a fine (i.e., segment the video into “smallest units” that are meaningful to them) or coarse-grain (i.e., segment the video into “largest units) of segmentation^{10,11}. Our instructions thus may have led to variation, even within individual, in how they approached the task (fine-grained versus coarse-grained) and this may have impacted reliability of SA.”

6. LL 427-439. As the authors noted, some of the videos have cinematography, they may be staged, and they are often edited (even documentaries can have staging and editing). This may introduce intentional and artificial segmentations. One way to address this in future work would be to measure context-based segmentation. Some of the videos used here were borrowed from the inferential emotion tracking paradigm, which had separate context-only, actor-only, and context+actor conditions created from the same videos. The cinematography is controlled in this way (e.g., comparing context versus actor) so any difference in segmentation would not be caused by staging but by differences in the availability of segmentation-relevant information. It is not necessary to conduct additional experiments for this manuscript. However, adding this to the future directions portion of this paragraph would be worthwhile because it helps address a potential concern readers will have.

We thank the reviewer for their suggestion. The reviewer is correct in noting the possible artifacts due to cinematography. We have already discussed this as a limitation in the Discussion section (page 15, lines 502-508):

“Furthermore, we did not quantify the possible impact of cinematography and editing on people’s segmentation of clips. In event segmentation, situational features in narrative films such as the discontinuity in time, space, or spatial movements and actions can predict the identification of event boundaries^{12,13}. Although our clips are shorter and often focus on a single spatiotemporal setting, there are cinematography details such as zooming in and switching between focus on faces and the broader situational scene, which may serve as cues for participants to identify emotional instances.”

We did not add the suggestion to compare blurring conditions since cuts may still impact segmentations across these conditions and the goal of our work is not to address the individual sources of information that perceivers are relying on (controlling for cuts), given that this question has already been addressed in a thorough manner in the inferential tracking tasks previously discussed. We have already included a suggestion for how to address the impact of cinematography in our discussion of future directions (page 16, lines 540-541):

“Future research may also benefit from coding cinematographic manipulations such as cuts and examining if the cinematographic pattern is correlated with the emotion segmentation.”

7. Fig. 4 The event segmentation suggests the videos in study 1 disproportionately featured anger, but how do we know this is not caused by the segmentation task itself? Perhaps segmentation is easier for anger? Study 2 videos partially address this, but anger still features quite prominently (the peak cell in Fig 4b is anger, and anger has one of the highest if not the highest average percent). The videos in Chen displayed a broader and more uniform range of emotions (both affect and categorical emotion) as measured by continuous tracking, but only a small subset of those videos was used here, so perhaps the video selection introduced the bias. Is there a way to measure whether anger disproportionately features in virtue of the segmentation paradigm itself?

The reviewer is correct at identifying the disproportionate inclusion of anger in Study 1 (indeed, we pointed this out in our manuscript on page 12, Figure 4 legend). To further document and investigate whether anger is disproportionately featured in the spontaneous segmentations of participants, we computed the frequency of generated emotion labels and obtained the top five most generated labels for each video to investigate the effect of anger on segmentation performance. For Study 1 stimuli, 4 out of 6 emotional videos have anger/angry in the top 5 labels. Among those 4 videos, 3 of them also have the highest average frequency of segmentation across subjects out of all videos. These results are now added to Supplementary note 7.

We have added a brief discussion of this to the newly added limitations section, page 15, lines 490-494.

“The current study has several limitations. For one, the stimuli in Study 1 disproportionally featured anger (Fig.4). It is possible that the number of anger segmentations observed are not entirely stimulus driven but based on the accessibility of the anger category for participants. We do observe that in videos where targets are described as “angry” there tend to be more segmentations, on average (see Supplementary note 7).”

8. Related to the above. Beyond just “anger” there seems to be a disproportionate prevalence of negative valence in Study 1 videos. Across many negative categories there is higher percent. A split half on positive vs negative affect would probably really highlight this. The source may be the higher salience of negative versus positive events. Additionally, there could also be a normalization problem: what if a video has an overall very positive

tone, but is punctuated by changes in emotion. If the mean emotion/affect is, on average, already quite positive/high, then the only salient emotion change there can be is in a negative direction. That will look like many negative segmentations, and imply that the videos contain a lot of anger. That would be incorrect, though: the video is on average very positive but the salient changes can only occur in a negative direction. Study 2 videos seem more balanced on this, but the interpretation of the segmentations remains a bit unclear.

Related to the above comment, we acknowledge our limitation at selecting the videos. We have revised the Discussion section to include discussion regarding this limitation (page 15, lines 496-501):

“Our standard of selecting videos based on dynamics of emotions featured may have inadvertently resulted in a disproportionate inclusion of videos featuring negative emotions such as anger. It is possible that segmentation for negative emotions is easier, hence skewing the performance of segmentation. This limitation can be addressed with replications and more data-driven or theoretically motivated clip selection.”

We have also added the following language to the future directions section (page 16, lines 533-539):

“Future research using an affectively balanced stimulus set would allow for additional insights into how segmentation behaviors vary by valence (or other affective features). For example, it may be that individuals engage in more fine-grained segmentation for negative compared to positive emotional stimuli. This would fit with prior literature suggesting a negativity bias^{14,15} and that negative emotion such as anger sharpens attention^{16,17}. This would also allow for investigation of clinical samples with documented differences in the perception of positive versus negative emotions^{18,19}.”

9. Minor: The “Emotion Segmentation Paradigm” is capitalized. The Empathic Accuracy paradigm” is partially capitalized. The “inferential emotion tracking task (paradigm)” is not capitalized. Either capitalize them all, or don’t, but be consistent. Also, be consistent about the use of the word “paradigm” or “task” since those words are interchangeable in all three of these cases.

We thank the reviewer for the editorial comment. We have capitalized all references of paradigms (ES, EA, IET) and used “paradigm” for the three cases.

10. Fig 3. The CEA-NUL correlation is very strong and interesting. What about the possibility that this is due to the threshold idea in point 4 above? That is, one may recognize the need to segment without having the words for it. In that case, a participant

might set a different criterion for pausing, but have the exact same sensitivity (MLU might change but that's an ambiguous metric). This isn't so much caused by an association between vocabulary and sensitivity-to-segmentation but between language and criterion-for-pausing-videos. A couple of ideas to address this are to play back videos segmented by high vocabulary participants (a kind of ground truth) to those with low language scores. Can those low-vocab individuals nevertheless recognize the correct placement of segmentations (versus, say, lure segmentations in the wrong moments)? This kind of sensitivity could be measured in many different ways, like recollection and familiarity, for example.

The reviewer asks if the link between CEA and NUL is driven by having a liberal bias, rather than sensitivity. In the instruction, we explicitly asked participants to segment even without having the word, but we acknowledge the possibility that people may have different criteria at segmenting due to other reasons. As mentioned above in comment 4, we conducted additional analyses where we compute SDT parameters and examine how they relate to our segmentation metrics and how these SDT metrics may account for the link between CEA and NUL.

Pertaining to comment 4, we examined the relationship between signal detection metrics (i.e., sensitivity and criterion) and the main metrics of interest (namely CEA, NUL and Age of Acquisition, AoA). We observed that for both samples, sensitivity is positively associated with our primary measure of emotion segmentation performance, greater CEA (Study 1: $\rho = .58$; Study 2: $\rho = .52$). This suggests that the CEA metric is, in part, reflecting the ability to distinguish events. However, the relationship between CEA and criterion is extremely strong (Study 1: $\rho = -.92$; Study 2: $\rho = -.87$), suggesting that consensus-based emotion segmentation performance is tracking closely with having a liberal bias to indicate that an emotion is present.

Similarly, NUL also negatively correlated with criterion (Study 1: $\rho = -.86$; Study 2: $\rho = -.87$), suggesting that there's an association between emotion vocabulary and criterion for pausing videos.

We have added these analyses to the supplementary materials and have added a brief discussion of these results on pages 7-8, lines 254-258 and page 10, lines 354-360 of the main text:

“Similar to the emotion segmentation metrics, we conducted exploratory analyses to examine the relationship between the active emotion vocabulary metrics and signal detection parameters of criterion and sensitivity. We observed that both NUL and AoA correlated only with criterion, but not with sensitivity. For additional details of these analyses, see Supplementary Note 6.”

“In the exploratory analyses examining the relationship between our metrics and signal detection metrics, we observed that the primary emotion segmentation metric, CEA, and both active emotion vocabulary metrics (i.e., NUL and AoA) correlated with segmentation criterion. To examine whether criterion plays a role in the link between CEA and active emotion vocabulary metrics, we also conducted multivariate regression analysis. Results revealed that neither active emotion vocabulary metric predicted CEA when controlling for criterion. For additional details of these analyses, see Supplementary Note 6.”

We also include an extended discussion of the links between language metrics and the new signal detection metrics quantifying segmentation performance in the Discussion section on page 14, lines 458-469.

“Exploratory analysis using signal detection metrics revealed that both primary emotion segmentation metric CEA and active emotion vocabulary metrics are correlated with a more liberal criterion to segment. People with a broader and more sophisticated active emotion vocabulary may achieve higher agreement with others by being liberal about what counts as an instance of a change in emotion. It is possible that the Emotion Segmentation Paradigm has encouraged a more liberal strategy, given how participants were instructed (i.e., to segment even when they do not immediately know what the emotional change is). Participants may be motivated by the task instruction to “identify” emotions regardless of actual presence of emotion. Prior research suggests that individuals may adjust their criterion to be more liberal if the structure of the environment makes it more costly to miss a target²⁰. While there was technically no cost to missing a target in our paradigm, participants may have interpreted the instructions in such a manner.”

We also add a future direction to further examine the exploratory findings on page 16, lines 554-557:

“Our exploratory analysis using signal detection metrics demonstrated a positive link between active emotion vocabulary and a liberal criterion for pausing videos. To probe the link between emotion vocabulary and sensitivity to emotion events future research could use a more classic detection task where the events (and foils) are paused for the participant.”

Comments from Reviewer #2

1. The authors propose a new framework for studying emotion expressions as events rather than objects. Studying emotion perception within a dynamic context and doing so using

free labeling is an interesting approach. However, it should be noted that this approach is not precisely new (as the authors note, it has been used in studies on empathic accuracy) nor does it in fact address pertinent problems in current emotion perception research (it may address problems in emotion perception research as it was conducted in the second half of the last century), see also below.

We thank the reviewer for the comment. We agree that the field has made significant progress in moving away from overly simplistic paradigms, including the increased usage of the Empathic Accuracy Paradigm and the newly introduced Inferential Emotion Tracking Paradigm, both of which focus on emotion perception from dynamic stimuli. We believe our paradigm draws on the strengths of these existing approaches to 1) capture the individual differences in the identification of emotion changes using a consensus-based approach, and 2) examine the role of emotion labels (and the conceptual knowledge they anchor) in emotion perception. We have discussed the differences and similarities between the paradigms in page 4, lines 119-127 in the revised manuscript (for more detailed discussion regarding IET, see response to Reviewer 1, comment 1):

“Not only do we examine consensus in emotion segmentations, similar to the Inferential Emotion Tracking Paradigm², but we also innovate by examining individual differences in adherence to this group level consensus^{21,22}. Consensus is a powerful approach because it reflects individual’s fit with their cultural group(s)^{23–26} and can be beneficial in domains where there is no clear “accuracy” criterion²⁷. Our focus on group consensus therefore also distinguishes the paradigm from the Empathic Accuracy Paradigm^{3,4}, which allows participants to indicate observed emotions, but focuses on emotional displays within dyadic or single narrator settings and evaluates performance based on agreement with the target person only.”

Despite this progress in the field, we believe that several issues persist in the current emotion perception literature, including reliance on an object perception framework, trial-based designs, as well as the introduction of pre-determined emotion labels. We discussed these limitations in page 2-3, lines 58-70:

“Previous research examining the contextual nature of emotion inference is still largely characterized by paradigms rooted in the object perception framework, in which researchers rely on trial-based designs with single cues as targets. This critique includes prior research focused on language as a form of top-down constraint. Provided emotion words in forced-choice designs may serve as guardrails or distractors that result in inflated or deflated performance compared to designs such as free-labeling. Extant paradigms that measure emotion perception dynamically (and thus not as static objects) have limited contextual cues (e.g., the Empathic Accuracy Paradigm^{3,4}) or introduce

emotion labels as response choice (e.g., the Inferential Emotion Tracking Paradigm²). These task features hence may limit our ability to observe individual variation in the active emotion vocabulary. Studying individual differences in the active emotion vocabulary may lend insight into how emotion perception proceeds outside of the confines of laboratory settings in more naturalistic and complex contexts. ”

2. The metrics used are also strongly biased towards departures from consensus (both in terms of emotion discrimination as in terms of word use) as the sole individual difference of interest. This limits its applicability for clinical and other contexts where it does not suffice to know that someone does not share the consensus view (which is often a given). Rather than knowing that individuals identified as X do not share consensus on anger expressions it would be more important to know how they perceive these expressions instead.

We do believe that our focus on departures from consensus does hold some clinical relevance as is. For example, we discussed how this paradigm may be especially well suited for studying populations with heightened alexithymia (page 16, lines 564-567). That said, we agree that departures from consensus can also be examined for particular content. For instance, normatively negative, positive and affectively ambiguous events would be interesting to distinguish between for studying segmentation performance of depressed individuals. We have added a future direction to this effect on page 16, lines 558-563:

“Our paradigm can be further extended to examine emotion segmentation in clinical populations. For example, past research suggests that individuals with depression have a bias toward content of negative valence, which may lead to different perceptual attention²⁸ and interpretation^{29,30} of emotional stimuli. Future research with depressed samples could distinguish between normatively positive, negative and ambiguous emotional events, which would allow for more precision in measuring what is driving departures from consensus.”

3. That said, studying emotions as events is certainly of interest. It is just that it does not provide a better account of emotion expression but rather a complementary account that may be useful for specific research questions for which emotion discrimination ability and consensus perception are of special relevance.

We agree that this work is complementary to prior approaches and have edited our language to soften our claims. For example, our discussion now states (page 13, lines 425-426):

“Our study complements prior empirical efforts^{1,2} by examining individual differences in dynamic emotion perception without introducing predetermined emotion categorical labels.”

Further, when we contrast our work to the object perception framework (which we believe still characterizes and motivates empirical emotion perception research), we have adjusted our language to point out that our work focuses on modeling how emotion inferences unfold in more naturalistic and complex contexts, rather than characterizing these approaches as “hindering” the study of emotion perception (see Reviewer 2, comment 1 for these quotes in context).

4. The main rationale for switching the framework for studying emotion expressions is predicated by a – not uncommon – confusion between emotion production and emotion perception. These are not the same process and as such issues that are relevant to emotion production do not have to be relevant to emotion perception. Thus, while it is the case that humans tend to not produce highly stereotypical facial expressions in everyday circumstances, this does not mean that humans do not perceive emotion expressions as reactions to singular events. The use of emotion expressions in the films and literature very nicely demonstrates that most human perceive emotion expressions as signals of internal states, i.e., emotions, even if it is not clear that they were produced that way.

While we agree that it is important not to confuse evidence for production with evidence for perception, low base rates of the production of certain emotional expressions are relevant for motivating the approach of this study, specifically the use of naturalistic video stimuli. We now discuss this in more details the Introduction (page 2, lines 38-48):

“...this object perception framework is motivated by the assumption that there is a robust mapping between facial actions and inner emotional experiences. Yet people only rarely generate predicted patterns of facial expressions during instances of emotion^{31,32}, and there is remarkable diversity in the range of facial cues that are used to convey the same emotion³³. For example, if there was robust mapping between the internal emotional state of anger and scowling, studying how people perceive isolated scowls would be sufficient to understand how people perceive anger. However, there are many instances when an individual could infer anger that are not captured by a paradigm where a scowling expression is presented to a perceiver. The empirical evidence does not imply that there is no signaling via the face or other channels in emotional episodes but instead implies that the signaling is complex and variable.”

Further, the reviewer is interpreting our findings as support for the idea that perceivers use emotional facial expressions to infer internal states. We do not think we can conclude this from

these data for two reasons. First, these stimuli are complex and a range of information (including both nonverbal cues, semantics from the dialogue and conceptual meaning from the situational context) may be driving emotion segmentation. Second, prior work by Chen and Whitney², using some overlapping stimuli, revealed that perceivers rely on the face only minimally to infer emotion, and that other contextual cues appear to be the primary drivers of emotion inferences. It is likely that these same patterns of cue usage would extend to our work (despite this not being the primary goal of our series of studies). As such, we have refrained from adding such an interpretation to our study. We have added the examination of the value of faces as a future direction (page 16, lines 542-544):

“In addition, building on the findings of prior work demonstrating context as a strong driver of emotion inferences², future research could also further examine whether particular cues such as facial movements or aspects of the context drive segmentations.”

5. The authors also grossly overstate the impact of context and prior labeling on emotion perceptions. In the cited study [10] only anger expressions in a disgust context were labelled as disgust rather than anger. The same expressions in sad or fear contexts were still labelled as anger. Given the well-known strong perceptual overlap between anger and disgust this study really only shows a modest contribution of context. The authors also seem to underestimate the proliferation of the use of dynamic and contextualized stimuli in more recent emotion research. In this sense they rightly criticize a research approach that hardly anyone still does anymore.

We agree that the study by Aviezer and colleagues cited does not itself provide evidence for contextual effects across many categories, but this does not undermine the larger point we are making that is summarizing the impact of a number of different forms of context on emotion perception. In terms of the impact of physical context on emotion perception, we have now added some additional citations to provide the reader with more empirical examples (page 2, lines 48-51).

“Second, perceivers appear to rely heavily on contextual cues when inferring emotion, including physical contexts like scenes³⁴ and what a target is doing with their body³⁵⁻³⁷, information about the unfolding situation^{1,33,38} and the meanings conveyed by what others say³⁹.”

It should also be noted that the impact of context on putatively canonical expressions likely underestimates the impact that context exerts in real world situations, where facial actions are more variable and ambiguous. Indeed, several of the studies we cited make this point. We believe it is beyond the scope and focus of this paper to further argue this point, however.

Regarding the shifts in the use of the “object perception” type paradigm, we agree that there has been movement in the literature in the incorporation of dynamic and contextualized stimuli, including in research we cited^{1,2,40,41}. This does not mean that the use of static facial portrayals is no longer represented in the literature, however. In the last year alone, the journal *Emotion* published 14 articles using stimulus sets of facial expressions⁴²⁻⁴⁴. Further, large-scale data collection efforts, such as the Adolescent Brain and Cognitive Development (ABCD) study⁴⁵, continue to use static emotion portrayals in tasks meant to probe “emotional processing”.

6. That available words constrain what can be said is a truism. However, this also applies to the proposed new framework. Hence it is not clear why the new framework presents an advantage on this count.

We do not introduce words to the task, thus this critique does not apply to our study. We argued that the emotion words introduced in the task (via instruction or forced-choice options) can impact emotion perception behaviors, as we’ve discussed in the Introduction (page 2, lines 65-70, cited above with discussion on existing paradigm introducing emotion words). Our paradigm circumvents this artifact by asking people to generate their own words, rather than imposing pre-determined emotion words.

We agree that the original phrasing did invoke a truism. As a result, we have adjusted to avoid this statement and instead to focus on the consequences of the presence of labels. Specifically, we now state (page 2-3, lines 66-70, see Reviewer 2, comment 1 for these quotes in context):

“These task features hence may limit our ability to observe individual variation in the active emotion vocabulary. Studying individual differences in the active emotion vocabulary may lend insight into how emotion perception proceeds outside of the confines of laboratory settings in more naturalistic and complex contexts.”

7. Given that “standardized emotion stimuli” are usually validated by means of assessing consensus ratings, the use of consensus to assess accuracy is also anything but new. The only difference seems to be that the authors assess consensus on the fly (within their experiment) whereas more typically the consensus was established first, and accuracy the represents – as it does here – the agreement or disagreement with this consensus.

Our paradigm, similar to Empathic Accuracy Paradigm, draws from labels spontaneously generated by the participants. We then utilized consensus-based methods to evaluate semantic agreement with the group consensus. This is different from standardized stimulus set, where consensus is applied during the validation of stimuli but still driven by matching between pre-determined emotion labels and facial expression stimuli. We have edited the manuscript to illustrate the difference discussed above in the Discussion section (page 14, lines 444-448)

“Many standardized emotion stimuli (e.g., NimStim⁴⁶; CEED⁴⁷) are validated through forced-choice tasks, where a separate sample of participants rate the matching between the stimuli (usually facial expressions) and pre-determined emotion labels. In contrast, our paradigm constructs group-level semantic consensus from labels generated by participants, and evaluates semantic agreement with the group-consensus. ”

8. It should also be noted that free labeling is not really a solution to the points with regard to the influence of language. First, people can only use the words they know and if they do not know the “correct” word, they will surely not be able to use it. Giving labels does not necessary distort the labeling process as such labels are usually commonly known words, the meaning of which most participants are likely to remember even if they would not produce the word spontaneously.

Free labeling is of interest precisely because individuals may have semantic knowledge about emotions, but may not access that knowledge in everyday instances of emotion perception. We use the term “active” emotion vocabulary for precisely this reason. Our study of emotion segmentation focuses on individual variation in the absence of task structure. It is a separate empirical question whether introducing labels would lead to more uniform emotion segmentation performance across individuals. We pose this as an interesting future direction on page 17, lines 582-588:

“Future research could focus on testing this predictive account more directly by manipulating the predictability of incoming sensory inputs in the form of facial expressions, vocal tone, and other verbal or non-verbal cues and examining the corresponding impact on emotion segmentation. *For example, it would be interesting to similarly constrain performance by introducing labels, thereby activating a similar emotion vocabulary across individuals, and examine the impact on performance relative to a control condition.*”

We also agree that one may have access to the semantic content of the emotion without spontaneously generating the word for it. In our instruction, we explicitly instructed participants to segment even when they did not have a specific word in mind, but we also acknowledge that this may have led people to adopt a different criterion for segmentations (for investigation of this point, see Reviewer 1, comment 4 and comment 10, and Supplementary Note 7).

9. More importantly, as already shown very nicely by Frijda (1953) when free-labelling people do not necessarily produce emotion words but very often focus on either the antecedents or the action tendencies associated with the expression. In a context, where emotions are viewed as valuable social signals knowing what produced the expression or

what behaviors to expect is arguably more relevant than simply affixing a label. This, however, makes it more difficult to assess consensus as some people may use a label (he is angry), others an antecedent (he looks like someone insulted him), and yet others an action tendency (he looks as if he wants to hit someone), all of which may or may not relate to the same emotion concept. The analyses procedure used the authors is mainly focused on quantifying the use of similar words rather than on actual context and meaning. It is hence unclear to what degree the approach can actually assess emotion decoding – the purported goal.

While we focused our primary linguistic analyses on emotion labels, we instructed participants to label the emotion in “words or phrases” to allow for just such responses that Frijda⁴⁸ first noted. We have added this justification to the methods section on page 18, lines 647-650.

“Participants were encouraged to use a phrase if they did not have access to an emotion label given previous work suggesting that people make meaning of emotional “expressions” in variable ways, including describing antecedents and action (tendencies)⁴⁸.”

While descriptions other than emotion labels are not the focus of the current study, the Emotion Segmentation Paradigm has the potential to be expanded to examine more heterogeneous descriptions of emotion in future work. We have revised the manuscript to reflect this potential as a future direction (page 16-17, lines 568-575):

“The Emotion Segmentation Paradigm can also be expanded in future work to more explicitly examine a broader range of labeling behaviors beyond emotion labels. A long-standing observation is that individuals label emotional targets with a range of terms including antecedent conditions and action tendencies⁴⁸, which can be considered constitutive aspects of emotional events⁴⁹. Further, cultural groups vary in the tendency to describe emotions using action verbs, bodily state descriptions and mental states descriptions⁵⁰⁻⁵². The Emotion Segmentation Paradigm would be a beneficial framework for examining whether these different ways of making meaning impact the grain and relative timing of emotion segmentations.”

11. The metrics used by the authors focus on temporal agreement. That is, to what degree to people agree on a change in observed behavior. This however refers to the ability to discriminate emotions, which is absolutely not the same as recognizing emotions. Only NUL even refers to the content of the labels and there also in a very reduced manner focusing on consensus only. Yet a focus on consensus is actually contrary to the stated rationale which aims to assess constructivist predictions as well. A constructivist prediction would accept that individuals construct meaning based on expression and context – and because this

construction is done by individuals' departures from consensus are as meaningful as is adherence to consensus.

The reviewer correctly identified that our most robust results focus on temporal agreement of emotion segmentation behaviors. Regarding the content of labels, in addition to NUL, in Section 2.2 and more specifically in Supplementary Note 3, we delineated the metrics Sentiment Agreement Score (SAS) and the Sentiment Distinctiveness Score (SDS). SAS, as we described, captured “the (graded) agreement between the sentiment expressed by the participant’s label (or phrase) and the average sentiment of the group using the AffectVec database”. SDS, using the same database, “captured the within-person differentiation between different emotion concepts.” Both metrics were constructed based on consensus events generated by the groups, and complement NUL as quantification of the content of the labels. These metrics bear the potential to quantify semantic agreement that reflect how people recognize emotions in consensus with each other, not just regarding “when” or “where”, but “what” the emotion is. The way we quantify emotion encoding is dependent hence on agreeing *about* labels - both the timing of emotion labeling and the semantics of the label. The “focus on similar word use” is a secondary aim, however. The exploratory analysis finding that these semantic metrics are not correlated with emotion segmentation metrics might indeed be attributed to individual departures in meaning making that are predicted by the constructionist framework. We have gone through our manuscript to ensure we do not use any language that implies that emotion segmentation is a measure of “recognition”. Indeed, the goal of this line of work is to examine emotion inferences in a distinct manner from the classic recognition work.

Comments from Reviewer #3

1. "We also focused our analysis on a subset of metrics ... followed our pre-registered plans (available here: https://aspredicted.org/see_one.php) " (line 152):

I cannot access the preregistration. To look up a preregistration on AsPredicted, I need the AsPredicted number and the last name of the/an author. I do not see the number.

We thank the reviewer for identifying this issue. We have updated the manuscript (page 6, line 169) to include a new link that should make the pre-registration accessible (This is the link, for convenience: https://aspredicted.org/84G_GQB).

2. "The age range and ethnic/racial identity of the people featured were also more diverse than those featured in Study 1 stimuli" (line 691):

It would be useful to know how specifically diverse the people were (e.g., apparent races, actors' races/ages).

We thank the reviewer for their comment. We have added the demographic information (apparent race, gender, age) of the people featured in Study 2 stimuli in Supplemental Note 1 (page 3, lines 57-65) as followed:

“...Among all the clips, 9 were deemed to be of high quality with respect to the guideline and controlling for reappearing characters. These include 2 clips from *Extremis* (2016) (77s, mainly featuring one middle-aged White female; 67s, mainly featuring one middle-aged Black male and one young South East Asian female), 1 clip from *Twinsters* (2015) (166s, mainly featuring two East Asian teenaged girls), 1 clip from *A Secret Love* (2020) (66s, mainly featuring two White males and two White females, all older adults), 1 clip from *Unrest* (2017) (91s, featuring one young Black male and one young Black female), 2 clips from *Minding the Gap* (2018) (74s, featuring one young White male and one young White female; 111s, featuring one Black teenaged boy), 1 clip from *Found* (2021) (94s, mainly featuring two East Asian teenaged girls), and 1 clip from *For Akheem* (2017) (123s, mainly featuring one teenaged Black male and one teenaged Black female).”

3. " On the level 486 of subjects, we also used the outliers_{mad} function($b = 1.4826$, threshold = 3) in the Routliers package to detect outliers for each of the five Emotion Segmentation Paradigm metrics, ..." (line 485)

It would be useful if you could at least conceptually explain how the outliers are determined using outliers_{mad} function? Since a lot of key points in the manuscript are about the consensus in emotion judgments and you do not want to appear as if you were removing people with idiosyncratic (yet valid) emotion perception pattern from further analysis.

We thank the reviewer for the comment. We have added the explanation under Methods section (page 17-18, lines 609-616) as below:

“On the level of subjects, we also used the outliers_{mad} function($b = 1.4826$, threshold = 3) in the Routliers package to detect outliers for each of the five Emotion Segmentation Paradigm metrics, the two lexical complexity metrics, as well as the four questionnaire scores (as described below). The outliers_{mad} function uses the Median Absolute Deviation Method to detect outliers based on deviation from the median of the residuals, ensuring robustness of detection for data that is not normally distributed. This resulted in 40 total data points being removed, spanning across 40 participants and 11 metrics.”

4. We merged consensus events that were within 800ms to account for the delay in reaction time to press pause⁷⁰. The timestamps of the merged consensus events were the average of the timestamps of the overlapping consensus events. (line 620)

Why 800ms? A rationale would be useful.

We thank the reviewer for the comment. We use 800ms as the threshold to account for individual differences in reaction time. Prior research suggests that critical events identified within 800ms may be attributed to individual differences in normative reaction time rather than reflecting actual group consensus on the emotional event⁵³. We have also updated the Method section (page 21, lines 757-760) to reflect this rationale as below:

“We merged consensus events that were within 800ms to allow for potential individual differences in reaction time to press pause⁵³. Critical events identified within 800ms may be attributed to individual differences in normative reaction time rather than reflecting actual group consensus on the emotion event.”

5. There are many acronyms. Some of them are introduced in the text without the full spelling. Even after a couple of reads, I find myself confused. This issue stood out to me in particular in figures; sometimes it has a label that I cannot fully understand even when I refer to the main manuscript (e.g., "SAS43" in Figure 2b). It would be useful to have acronyms written at the end of every figure legend for those that are not familiar with the paradigm.

We thank the reviewer for the comment. We have updated the figure legends as below to include the full spelling corresponding to each acronym:

Figure 2. Correlation within metrics and between metrics and established measures. **a,** The emotion segmentation metrics, Mean Length of Unit (MLU), Consensus Event Agreement (CEA) and Segmentation Agreement (SA), all significantly correlated to one another. **b,** The active emotion label metrics, Number of Unique Labels (NUL), Semantic Agreement Score (SAS), Age of Acquisition (AoA) and Familiarity correlated with each other to varying degrees...For an equivalent figure for Study 2, see Supplementary Figure 5.

Figure 3. Correlation between emotion segmentation metric and active emotion vocabulary metric in Study 2. The primary emotion segmentation metric Consensus Event Agreement (CEA) is positively correlated with both active emotion vocabulary metrics, Number of Unique Labels (NUL) and Age of Acquisition (AoA), suggesting that people with better segmentation performance also possess larger and more advanced emotion vocabularies. See Supplementary Figure 6 for an equivalent figure in Study 1.

6. It would be useful to have an equivalent figure to Figure 2 for study 2 as well. That will help the audience gauge the rigor of the main effects found here.

We thank the reviewer for the comment. We have generated the corresponding figure as Supplementary Figure 5 (also included here) and referred the reader to Supplementary Material in the figure legends of Figure 2 (see above).

Supplementary Figure 5. Correlation within metrics and between metrics and established measures in Study 2. a, While the emotion segmentation metric Consensus Event Agreement (CEA) significantly correlated to both Mean Length of Unit (MLU) and Segmentation Agreement (SA), the latter two metrics did not correlate with each other. b, While the two active emotion vocabulary metrics Number of Unique Labels (NUL) and Age of Acquisition (AoA) significantly correlated with each other, neither correlated with Semantic Agreement Score (SAS). c,d, While three paradigm metrics (SA, CEA, NUL) correlated with both GERT-S scores and STEU-B scores in the expected direction, NUL was not correlated with either. The graph is the scatterplot of ranked data.

7. Likewise, it would be useful to see Figure 3 equivalent for study 1. (2.4 The Link Between Active Emotion Vocabulary and Emotion Segmentation) I believe the effect was repeated for CEA.

We thank the reviewer for the comment. We have generated the corresponding figure as Supplementary Figure 6 (also included here) and referred the reader to Supplementary Material in the figure legends of Figure 3 (see above).

Supplementary Figure 6. Correlation between emotion segmentation metric and active emotion vocabulary metric in Study 1. The primary emotion segmentation metric Consensus Event Agreement (CEA) is positively correlated with both active emotion vocabulary metrics, Number of Unique Labels (NUL) and Age of Acquisition (AoA) (NUL, AoA), suggesting that people with better segmentation performance also possess larger and more advanced emotion vocabularies. The effect was replicated in Study 2.

8. The way why Positive Well Being Scale and Autism Spectrum Quotient were included in the questionnaire series was too ambiguous in my opinion ("We were specifically interested in the Positive Relations with Others subscale ... and used this subscale score for subsequent concurrent validity testing."). It might be useful to briefly describe the respective prediction (what positive or negative relationship you predicted, between each of these measures and the measure from your paradigm)—just one short sentence each perhaps.

We thank the reviewer for the comment. We have added brief descriptions of each prediction under Methods section (page 19, lines 687-690; lines 698-700) as below:

“3) Psychological Well-Being Scale (PWB) is a 42-item scale that measures six aspects of wellbeing and happiness... We were specifically interested in the Positive Relations with Others subscale (Study 2: $\omega_T = 0.84$; Study 2: $\omega_T = 0.82$) and used this subscale score for subsequent concurrent validity testing. We expected that dynamic emotion perception should be associated with positive social relationships⁵⁴, reflected as a positive correlation between the Emotion Segmentation Paradigm metrics (described below in Data Analysis section) and the Positive Relations with Others subscale scores. ... 4) Autism Spectrum Quotient - 10 Items (AQ-10) is a 10-item scale commonly used as a screening tool for Autism Spectrum... This measure was included to allow for a robustness check, as we expect the metrics to demonstrate high reliability within non-autistic subsample who scored less than 6 on this screener tool. More details are further described in Supplementary Note 4.”

9. "These RSA metrics ..." (line 574) this may be confusing to the readers unfamiliar with RSA (also, the acronym should be spelled out previously) because what RSA is and how it is applied in this context is not clear. It would be useful to either (1) briefly explain how it was done, or (2) completely lose the acronym ("RSA") so the authors as well as readers do not have to worry about it. Either case, you could simply refer to Suppl Note 5.

We thank the reviewer for the comment. We have removed the term RSA and referred the reader to Suppl Note 5 (page 20, lines 712-714) as below:

...These metrics did not demonstrate significant association with the emotion segmentation metrics and were hence not further pursued in Study 2. Detailed description of the task, analysis and results can be found in Supplementary Note 5.

References

1. Chen, Z. & Whitney, D. Tracking the affective state of unseen persons. *Proc. Natl. Acad. Sci.* **116**, 7559–7564 (2019).
2. Chen, Z. & Whitney, D. Inferential emotion tracking (IET) reveals the critical role of context in emotion recognition. *Emot. Wash. DC* (2020) doi:10.1037/emo0000934.
3. Ickes, W. Empathic Accuracy. *J. Pers.* **61**, 587–610 (1993).
4. Ta, V. P. & Ickes, W. Empathic accuracy. in *The Routledge Handbook of Philosophy of Empathy* (Routledge, 2017).
5. Macmillan, N. A. & Creelman, C. D. *Detection Theory: A User's Guide*. (Taylor & Francis Group, 2004).
6. Hautus, M. J. Corrections for extreme proportions and their biasing effects on estimated values of d' . *Behav. Res. Methods Instrum. Comput.* **27**, 46–51 (1995).
7. Zacks, J. M. Event Perception and Memory. *Annu. Rev. Psychol.* **71**, 165–191 (2020).
8. Kurby, C. A. & Zacks, J. M. Age differences in the perception of hierarchical structure in events. *Mem. Cognit.* **39**, 75–91 (2011).
9. Sasmita, K. & Swallow, K. M. Measuring event segmentation: An investigation into the stability of event boundary agreement across groups. *Behav. Res. Methods* (2022) doi:10.3758/s13428-022-01832-5.
10. Newtonson, D. Attribution and the unit of perception of ongoing behavior. *J. Pers. Soc. Psychol.* **28**, 28–38 (1973).
11. Swallow, K. M., Kemp, J. T. & Candan Simsek, A. The role of perspective in event segmentation. *Cognition* **177**, 249–262 (2018).
12. Magliano, J. P., Miller, J. & Zwaan, R. A. Indexing space and time in film understanding. *Appl. Cogn. Psychol.* **15**, 533–545 (2001).
13. Magliano, J. P. & Zacks, J. M. The Impact of Continuity Editing in Narrative Film on Event Segmentation. *Cogn. Sci.* **35**, 1489–1517 (2011).
14. Vaish, A., Grossmann, T. & Woodward, A. Not all emotions are created equal: The negativity bias in social-emotional development. *Psychol. Bull.* **134**, 383–403 (2008).
15. Norris, C. J. The negativity bias, revisited: Evidence from neuroscience measures and an individual differences approach. *Soc. Neurosci.* **16**, 68–82 (2021).
16. Feldmann-Wüstefeld, T., Schmidt-Daffy, M. & Schubö, A. Neural evidence for the threat detection advantage: Differential attention allocation to angry and happy faces. *Psychophysiology* **48**, 697–707 (2011).
17. Ince, S. *et al.* Subcortical contributions to salience network functioning during negative emotional processing. *NeuroImage* **270**, 119964 (2023).
18. Dalili, M. N., Penton-Voak, I. S., Harmer, C. J. & Munafò, M. R. Meta-analysis of emotion recognition deficits in major depressive disorder. *Psychol. Med.* **45**, 1135–1144 (2015).

19. Wenzler, S. *et al.* Intensified emotion perception in depression: Differences in physiological arousal and subjective perceptions. *Psychiatry Res.* **253**, 303–310 (2017).
20. Lynn, S. K. & Barrett, L. F. “UTILIZING” SIGNAL DETECTION THEORY. *Psychol. Sci.* **25**, 1663–1673 (2014).
21. Geher, G., Warner, R. M. & Brown, A. S. Predictive validity of the emotional accuracy research scale. *Intelligence* **29**, 373–388 (2001).
22. MacCann, C., Roberts, R. D., Matthews, G. & Zeidner, M. Consensus scoring and empirical option weighting of performance-based Emotional Intelligence (EI) tests. *Personal. Individ. Differ.* **36**, 645–662 (2004).
23. Weller, S. C. Cultural Consensus Theory: Applications and Frequently Asked Questions. *Field Methods* **19**, 339–368 (2007).
24. Batchelder, W. H., Anders, R. & Oravecz, Z. Cultural Consensus Theory. in *Stevens’ Handbook of Experimental Psychology and Cognitive Neuroscience* 1–64 (John Wiley & Sons, Ltd, 2018). doi:10.1002/9781119170174.epcn506.
25. Mesquita, B. Emotions as dynamic cultural phenomena. in *Handbook of affective sciences* 871–890 (Oxford University Press, 2003).
26. De Leersnyder, J., Mesquita, B. & Kim, H. S. Where do my emotions belong? A study of immigrants’ emotional acculturation. *Pers. Soc. Psychol. Bull.* **37**, 451–463 (2011).
27. Legree, P. J., Psotka, J., Tremble, T. R. & Bourne, D. *Applying Consensus-Based Measurement to the Assessment of Emerging Domains*. <https://apps.dtic.mil/sti/citations/ADA430810> (2005).
28. Mennen, A. C., Norman, K. A. & Turk-Browne, N. B. Attentional bias in depression: understanding mechanisms to improve training and treatment. *Curr. Opin. Psychol.* **29**, 266–273 (2019).
29. Raes, F., Hermans, D. & Williams, J. M. G. Negative Bias in the Perception of Others’ Facial Emotional Expressions in Major Depression: The Role of Depressive Rumination. *J. Nerv. Ment. Dis.* **194**, 796 (2006).
30. Neta, M. & Whalen, P. J. The Primacy of Negative Interpretations When Resolving the Valence of Ambiguous Facial Expressions. *Psychol. Sci.* **21**, 901–907 (2010).
31. Barrett, L. F., Adolphs, R., Marsella, S., Martinez, A. M. & Pollak, S. D. Emotional Expressions Reconsidered: Challenges to Inferring Emotion From Human Facial Movements. *Psychol. Sci. Public Interest* **20**, 1–68 (2019).
32. Durán, J. I. & Fernández-Dols, J.-M. Do emotions result in their predicted facial expressions? A meta-analysis of studies on the co-occurrence of expression and emotion. *Emot. Wash. DC* **21**, 1550–1569 (2021).
33. Le Mau, T. *et al.* Professional actors demonstrate variability, not stereotypical expressions, when portraying emotional states in photographs. *Nat. Commun.* **12**, 5037 (2021).
34. Ngo, N. & Isaacowitz, D. M. Use of context in emotion perception: The role of top-down control, cue type, and perceiver’s age. *Emotion* **15**, 292–302 (2015).

35. Aviezer, H. *et al.* Angry, Disgusted, or Afraid?: Studies on the Malleability of Emotion Perception. *Psychol. Sci.* **19**, 724–732 (2008).
36. Aviezer, H., Trope, Y. & Todorov, A. Body Cues, Not Facial Expressions, Discriminate Between Intense Positive and Negative Emotions. *Science* **338**, 1225–1229 (2012).
37. Van den Stock, J., Righart, R. & de Gelder, B. Body expressions influence recognition of emotions in the face and voice. *Emot. Wash. DC* **7**, 487–494 (2007).
38. Ong, D. C., Zaki, J. & Goodman, N. D. Affective cognition: Exploring lay theories of emotion. *Cognition* **143**, 141–162 (2015).
39. Hodges, S. D. & Kezer, M. It Is Hard to Read Minds without Words: Cues to Use to Achieve Empathic Accuracy. *J. Intell.* **9**, 27 (2021).
40. Calbi, M. *et al.* How Context Influences Our Perception of Emotional Faces: A Behavioral Study on the Kuleshov Effect. *Front. Psychol.* **8**, (2017).
41. Barratt, D., Rédei, A. C., Innes-Ker, Å. & van de Weijer, J. Does the Kuleshov Effect Really Exist? Revisiting a Classic Film Experiment on Facial Expressions and Emotional Contexts. *Perception* **45**, 847–874 (2016).
42. Thorstenson, C. A., McPhetres, J., Pazda, A. D. & Young, S. G. The role of facial coloration in emotion disambiguation. *Emotion* **22**, 1604–1613 (2022).
43. Yitzhak, N., Pertzov, Y., Guy, N. & Aviezer, H. Many ways to see your feelings: Successful facial expression recognition occurs with diverse patterns of fixation distributions. *Emot. Wash. DC* **22**, 844–860 (2022).
44. Miller, E. J., Krumhuber, E. G. & Dawel, A. Observers perceive the Duchenne marker as signaling only intensity for sad expressions, not genuine emotion. *Emot. Wash. DC* **22**, 907–919 (2022).
45. Smolker, H. R. *et al.* The Emotional Word-Emotional Face Stroop task in the ABCD study: Psychometric validation and associations with measures of cognition and psychopathology. *Dev. Cogn. Neurosci.* **53**, 101054 (2022).
46. Tottenham, N. *et al.* The NimStim set of facial expressions: Judgments from untrained research participants. *Psychiatry Res.* **168**, 242–249 (2009).
47. Benda, M. S. & Scherf, K. S. The Complex Emotion Expression Database: A validated stimulus set of trained actors. *PloS One* **15**, e0228248 (2020).
48. Frijda, N. H. The understanding of facial expression of emotion. *Acta Psychol. (Amst.)* **9**, 294–362 (1953).
49. Ortony, A. & Clore, G. L. Emotions, moods, and conscious awareness: Comment on Johnson-Laird and Oatley’s ‘The language of emotions: An analysis of a semantic field.’ *Cogn. Emot.* **3**, 125–169 (1989).
50. Semin, G. R., Görts, C. A., Nandram, S. & Semin-Goossens, A. Cultural perspectives on the linguistic representation of emotion and emotion events. *Cogn. Emot.* **16**, 11–28 (2002).
51. Choi, E., Chentsova-Dutton, Y. & Parrott, W. G. The Effectiveness of Somatization in Communicating Distress in Korean and American Cultural Contexts. *Front. Psychol.* **7**, (2016).

52. Gendron, M., Crivelli, C. & Barrett, L. F. Universality Reconsidered: Diversity in Making Meaning of Facial Expressions. *Curr. Dir. Psychol. Sci.* **27**, 211–219 (2018).
53. Meitz, T. G. K., Meyerhoff, H. S. & Huff, M. Event related message processing: perceiving and remembering changes in films with and without soundtrack. *Media Psychol.* **23**, 733–763 (2020).
54. Zaki, J. Integrating Empathy and Interpersonal Emotion Regulation. *Annu. Rev. Psychol.* **71**, 517–540 (2020).

4th Aug 23

Dear Dr. Gendron,

Your manuscript titled "Emotional Event Perception is Related to Lexical Complexity and Emotion Knowledge" has now been seen by our reviewers, whose comments appear below. In light of their advice I am delighted to say that we are happy, in principle, to publish a suitably revised version in Communications Psychology under the open access CC BY license (Creative Commons Attribution v4.0 International License).

We therefore invite you to revise your paper one last time to address the remaining concerns of our reviewers and a list of editorial requests. At the same time we ask that you edit your manuscript to comply with our format requirements and to maximise the accessibility and therefore the impact of your work.

EDITORIAL REQUESTS:

Please address the remaining comments from Reviewer 2 and discuss this limitation.

SUBMISSION INFORMATION:

OPEN ACCESS:

Communications Psychology is a fully open access journal. Articles are made freely accessible on publication under a [CC BY](http://creativecommons.org/licenses/by/4.0) license (Creative Commons Attribution 4.0 International License). This license allows maximum dissemination and re-use of open access materials and is preferred by many research funding bodies.

For further information about article processing charges, open access funding, and advice and support from Nature Research, please visit <https://www.nature.com/commspsychol/article-processing-charges>

At acceptance, you will be provided with instructions for completing this CC BY license on behalf of

all authors. This grants us the necessary permissions to publish your paper. Additionally, you will be asked to declare that all required third party permissions have been obtained, and to provide billing information in order to pay the article-processing charge (APC).

* **DATA AVAILABILITY:**

[link redacted]

Best regards,

Jennifer Bellingtier

Jennifer Bellingtier, PhD
Senior Editor
Communications Psychology

REVIEWERS' EXPERTISE:

Reviewer #1 social/affective psychology; individual differences
Reviewer #2 social/affective psychology; emotion expression
Reviewer #3 social/affective psychology; event segmentation

REVIEWERS' COMMENTS:

Reviewer #1 (Remarks to the Author):

The authors have addressed all of my concerns. Nice paper.

Reviewer #2 (Remarks to the Author):

The authors revised manuscript addresses many of the points raised by me and the other reviewers. However, I still think that the claims to novelty need to be toned down in line with my original suggestions.

Also, the authors seem to have misunderstood my point 9. Given that it is well established that people use not only labels but also antecedents and action tendencies when free-labeling emotion expressions, the focus on labels only means that the authors underestimate emotion recognition. This may not be as obvious given that their metrics do not in fact relate to emotion recognition per se but rather to consensus but it remains a problem as it also underestimates consensus. For example, when one person refers to anger and another to the desire to hit someone then they may in fact express a consensus view (or not, which is precisely the problem that I alluded to).

Overall, I still feel that the approach is of very limited applicability as it can only be applied to research questions with a very small focus on consensus.

Reviewer #3 (Remarks to the Author):

The authors addressed my comments. I commend the authors. I have no further comments.

Response to Referees

Note: New text in provided quotes is in blue font and old text is in black font.

Comments from Reviewer #2

1. The authors revised manuscript addresses many of the points raised by me and the other reviewers. However, I still think that the claims to novelty need to be toned down in line with my original suggestions.

We thank the reviewer for their comment. We have further tempered the language to acknowledge previous work and tone down the claims to novelty (for example, on page 13, line 425, line 429, as shown below).

“Our study complements and **builds on** prior empirical efforts^{1,2} by examining individual differences in dynamic emotion perception without introducing predetermined emotion categorical labels.”

“Our approach **borrow**s a number of elements from previous work (i.e., the Inferential Affective Tracking¹ and Inferential Emotion Tracking Paradigms²), which assessed intersubject agreement in affect ratings and discrete emotion judgments from film and documentary clips.”

2. Also, the authors seem to have misunderstood my point 9. Given that it is well established that people use not only labels but also antecedents and action tendencies when free-labeling emotion expressions, the focus on labels only means that the authors underestimate emotion recognition. This may not be as obvious given that their metrics do not in fact relate to emotion recognition per se but rather to consensus but it remains a problem as it also underestimates consensus. For example, when one person refers to anger and another to the desire to hit someone then they may in fact express a consensus view (or not, which is precisely the problem that I alluded to).

We appreciate the opportunity to further clarify that our semantic consensus approach did not exclude non-emotion labels, only our measure of emotion label complexity. For example, if the word “rant” was used by a participant, this would load on the category of anger based on the affectvec word embedding space that we used. We have added this example to the supplementary material to clarify this point (page 17, 237-242).

“This semantic consensus approach also includes non-emotion labels generated by participants. For instance, if a participant generates the word “rant,” based on the

AffectVec embedding, the word will have the highest loading score on the emotion category of anger (0.4049), among other categories. The word will hence be considered in consensus with words that directly refer to anger (e.g., “angry,” which loads on anger at 0.7573).”

We also acknowledge that focusing on individual differences in vocabulary complexity for emotion labels is limited. We have further revised the Future Direction Section to reflect the potential of other ways of examining emotion inference with language (e.g., action tendencies) (page 17, lines 580-589).

“The Emotion Segmentation Paradigm can also be expanded in future work to more explicitly examine a broader range of labeling behaviors beyond emotion labels. A long-standing observation is that individuals label emotional targets with a range of terms including antecedent conditions and action tendencies¹¹, which can be considered constitutive aspects of emotional events¹². As a result, our focus on emotion labels when we examined individual differences in language complexity may not fully capture the range of concepts that structure emotion inference. Indeed, cultural groups vary in the tendency to describe emotions using action verbs, bodily state descriptions and mental states descriptions . The Emotion Segmentation Paradigm would be a beneficial framework for examining whether these different ways of making meaning (e.g., emotion labels versus action tendencies) impact the grain and relative timing of emotion segmentations.”

Overall, I still feel that the approach is of very limited applicability as it can only be applied to research questions with a very small focus on consensus.

We agree that the consensus-based approach can be limited in capturing “emotion recognition” as studied by prior work. We have revised the Limitation section to include this limitation (page 16, lines 532-543).

“Finally, a consensus-based approach used here is limited in that it does not capture other forms of “accuracy” that have been examined in the prior literature studying emotion “recognition”. Whereas some recognition studies use normative ratings of stimuli as ground truth and are thus ultimately relying on consensus, other approaches may use other ground truth criteria. Ground truth may be based on researcher-stipulated stimuli such as facial portrayals that contain specific muscle group movements^{3,4}, carrying assumptions about clear mappings between these cues and emotions. Ground truth may also be based on the situation in which the cues were elicited^{5,6}, carrying assumptions about clear mappings between situations and emotions. Finally, ground truth may be

based on the self-reported emotion of the target individual^{7–10} carrying assumptions about individuals' ability and motivation to access and label their emotional states. Such, each of these approaches has different strengths and weaknesses that should be weighed carefully.”

References

1. Chen, Z. & Whitney, D. Tracking the affective state of unseen persons. *Proc. Natl. Acad. Sci.* **116**, 7559–7564 (2019).
2. Chen, Z. & Whitney, D. Inferential emotion tracking (IET) reveals the critical role of context in emotion recognition. *Emot. Wash. DC* (2020) doi:10.1037/emo0000934.
3. Condliffe, O. & Maratos, F. A. Can compassion, happiness and sympathetic concern be differentiated on the basis of facial expression? *Cogn. Emot.* **34**, 1395–1407 (2020).
4. Clark, E. A. *et al.* The Facial Action Coding System for Characterization of Human Affective Response to Consumer Product-Based Stimuli: A Systematic Review. *Front. Psychol.* **11**, (2020).
5. Le Mau, T. *et al.* Professional actors demonstrate variability, not stereotypical expressions, when portraying emotional states in photographs. *Nat. Commun.* **12**, 5037 (2021).
6. Cowen, A. S. *et al.* Sixteen facial expressions occur in similar contexts worldwide. *Nature* **589**, 251–257 (2021).
7. Zaki, J., Bolger, N. & Ochsner, K. It Takes Two: The Interpersonal Nature of Empathic Accuracy. *Psychol. Sci.* **19**, 399–404 (2008).
8. Zerwas, F. K. *et al.* “I feel you”: Greater linkage between friends' physiological responses and emotional experience is associated with greater empathic accuracy. *Biol. Psychol.* **161**, 108079 (2021).
9. Jospe, K. *et al.* The contribution of linguistic and visual cues to physiological synchrony and empathic accuracy. *Cortex* **132**, 296–308 (2020).
10. Fujiwara, K. & Daibo, I. Empathic accuracy and interpersonal coordination: behavior matching can enhance accuracy but interactional synchrony may not. *J. Soc. Psychol.* **0**, 1–18 (2021).
11. Frijda, N. H. The understanding of facial expression of emotion. *Acta Psychol. (Amst.)* **9**, 294–362 (1953).
12. Ortony, A. & Clore, G. L. Emotions, moods, and conscious awareness: Comment on Johnson-Laird and Oatley's 'The language of emotions: An analysis of a semantic field.' *Cogn. Emot.* **3**, 125–169 (1989).
13. Semin, G. R., Görts, C. A., Nandram, S. & Semin-Goossens, A. Cultural perspectives on the linguistic representation of emotion and emotion events. *Cogn. Emot.* **16**, 11–28 (2002).

14. Choi, E., Chentsova-Dutton, Y. & Parrott, W. G. The Effectiveness of Somatization in Communicating Distress in Korean and American Cultural Contexts. *Front. Psychol.* **7**, (2016).